# Genome of the early spider-orchid *Ophrys sphegodes* provides insights into sexual deception and pollinator adaptation

Alessia Russo [1,2,3] ✉, Mattia Alessandrini[1], Moaine El Baidouri [4,5,6], Daniel Frei [7], Teresa Rosa Galise [8], Lara Gaidusch[1], Hannah F. Oertel [1], Sara E. Garcia Morales [1], Giacomo Potente [3], Qin Tian [9,10], Dmitry Smetanin[2], Joris A. M. Bertrand[4,5,6], Renske E. Onstein [9,11], Olivier Panaud [4,5,6], Jürg E. Frey [7], Salvatore Cozzolino[8], Thomas Wicker [2], Shuqing Xu [12], Ueli Grossniklaus [2] & Philipp M. Schlüter [1,3] ✉

Pollinator-driven evolution of floral traits is thought to be a major driver of angiosperm speciation and diversification. *Ophrys* orchids mimic female insects to lure male pollinators into pseudocopulation. This strategy, called sexual deception, is species-specific, thereby providing strong premating reproductive isolation. Identifying the genomic architecture underlying pollinator adaptation and speciation may shed light on the mechanisms of angiosperm diversification. Here, we report the 5.2 Gb chromosome-scale genome sequence of *Ophrys sphegodes*. We find evidence for transposable element expansion that preceded the radiation of the *O. sphegodes* group, and for gene duplication having contributed to the evolution of chemical mimicry. We report a highly differentiated genomic candidate region for pollinator-mediated evolution on chromosome 2. The *Ophrys* genome will prove useful for investigations into the repeated evolution of sexual deception, pollinator adaptation and the genomic architectures that facilitate evolutionary radiations.

Understanding the genetic mechanisms of adaptation to pollinators is a central question in plant evolutionary biology. The prominent role of pollinators in flower evolution is related to their dual function: enabling sexual reproduction and imposing selection on floral traits[1,2]. This is particularly evident in plant species with a specialised pollination mechanism associated with floral traits that evolved to attract one or a few functionally alike pollinator species[3,4]. Therefore, pollinator-mediated evolution of floral traits is considered a major force driving angiosperm diversity by contributing to their radiation[5,6]. It has been hypothesised that radiations are facilitated by entering a new ecological niche with little or no competition from similar species (ecological opportunity), and the genetic potential allowing the necessary

[1]Department of Plant Evolutionary Biology, Institute of Biology, University of Hohenheim, Stuttgart, Germany. [2]Department of Plant and Microbial Biology and Zürich-Basel Plant Science Centre, University of Zurich, Zürich, Switzerland. [3]Department of Systematic and Evolutionary Botany and Zürich-Basel Plant Science Centre, University of Zurich, Zürich, Switzerland. [4]Université Perpignan Via Domitia, Laboratoire Génome et Développement des Plantes, UMR5096 Perpignan, France. [5]CNRS, Laboratoire Génome et Développement des Plantes, UMR5096 Perpignan, France. [6]EMR269 MANGO, Institut de Recherche pour le Développement, Perpignan, France. [7]Department of Methods Development and Analytics, Agroscope, Wädenswil, Switzerland. [8]Department of Biology, University of Naples Federico II, Naples, Italy. [9]Naturalis Biodiversity Centre, Leiden, The Netherlands. [10]Germplasm Bank of Wild Species, Kunming Institute of Botany, Chinese Academy of Sciences, Kunming, China. [11]German Centre for Integrative Biodiversity Research (iDiv) Halle – Jena – Leipzig, Leipzig, Germany. [12]Institute of Organismic and Molecular Evolution, University of Mainz, Mainz, Germany. ✉e-mail: alessia.russo@botinst.uzh.ch; philipp.schlueter@uni-hohenheim.de

adaptations to evolve (genetic variation), ultimately leading to rapid bursts of speciation[7–9].

Orchids of the Euro-Mediterranean genus *Ophrys* ensure reproduction through sexual deception. Specifically, they mimic the olfactory, visual and tactile signals of females of their pollinating insects to entice conspecific males to pseudocopulate with the flower, leading to pollination (Fig. 1a, b). Among the flower traits adapted to pollinators, olfactory signals are pivotal to specific pollinator attraction[10,11], with selection by pollinators[12] leading to strong odour differentiation among closely related species[13]. Additional adaptations to pollinators likely involve flower labellum colour, which matches pollinator body coloration, and floral morphology that optimises pollen transfer[13]. At the same time, conspicuous UV-reflective patterns and odour compounds not primarily required for sexual attraction are highly variable between plants and likely aid male pollinators in memorising and avoiding plants, thereby increasing outcrossing rates[13–15]. This extreme specialisation of *Ophrys* floral traits makes the plant-insect interaction highly species-specific, and the lack of shared pollinators between species leads to strong premating reproductive isolation[12,13]. Notwithstanding some uncertainty about the number of species and the extent of pollinator sharing[16,17], at the local population level extreme pollinator specialisation is evident in *Ophrys* species[12,13,18]. The genus *Ophrys* is of relatively recent origin (c. 4.9 Ma crown age), with the earliest-diverged *Ophrys* lineage likely having been wasp-pollinated[19,20]. Two independent *Ophrys* lineages later experienced a pollinator shift to *Andrena* bee pollinators, preceding a burst of speciation in the *O. sphegodes* group within the last million years[19]. This resulted in a species radiation with one of the highest reported diversification rates among angiosperms[19]. While several facets of *Ophrys* speciation are relatively well understood[12,13,19], the genomic features allowing its adaptation to diverse pollinators are largely unknown. Our ability to elucidate how genome architecture has contributed to *Ophrys*

diversification and adaptation to pollinators has hitherto been hampered by the lack of a reference genome.

Here, we present the chromosome-level genome assembly of *Ophrys sphegodes*, a key representative of the genus and the adaptive radiation, to address questions on the genetic mechanisms underlying such rapid pollinator-driven speciation.

## Results and discussion
### Chromosome-level genome assembly

The orchid *Ophrys sphegodes* MILL. has a karyotype of $2n = 2x = 36$ [12,21,22]. The haploid genome size was estimated to be 4.83 Gb by flow cytometry, and heterozygosity was estimated at 1.28% via *k*-mer analysis (Supplementary Figs. 1 and 2, Supplementary Table 1). To assemble the genome, we generated a total of 409 Gb data on the Nanopore PromethION platform (Supplementary Table 2, Supplementary Fig. 3). Additionally, whole-genome Illumina sequencing data (WGS; 268 Gb, Supplementary Table 3) and Hi-C chromatin conformation capture libraries (Illumina, Supplementary Table 3) were produced to perform polishing and anchoring of scaffolds, respectively. We used Miniasm assembly[23] to generate a total of 11,148 contigs in 6.4 Gb, with an N50 value of 754 kb (Supplementary Table 4, Supplementary Fig. 4). We removed under-collapsed heterozygous contigs (1.2 Gb) and used Hi-C data to anchor the scaffolds into 18 pseudomolecules corresponding to the 18 chromosomes expected for the haploid genome (Fig. 1c, Supplementary Fig. 5a; Supplementary Table 5). The final assembled genome size was 5.2 Gb, in line with the estimated genome size, with a scaffold N50 of 218 Mb, L50 of 10 and L75 of 17 (Supplementary Table 4). Overall, 97.8% of raw Illumina WGS reads and 98.1% of an independent Pacific Biosciences (PacBio) dataset (not used for assembly; Supplementary Table 6) could be mapped to the assembly, suggesting that our assembly contains the complete genetic information. Gene region completeness was assessed using BUSCO

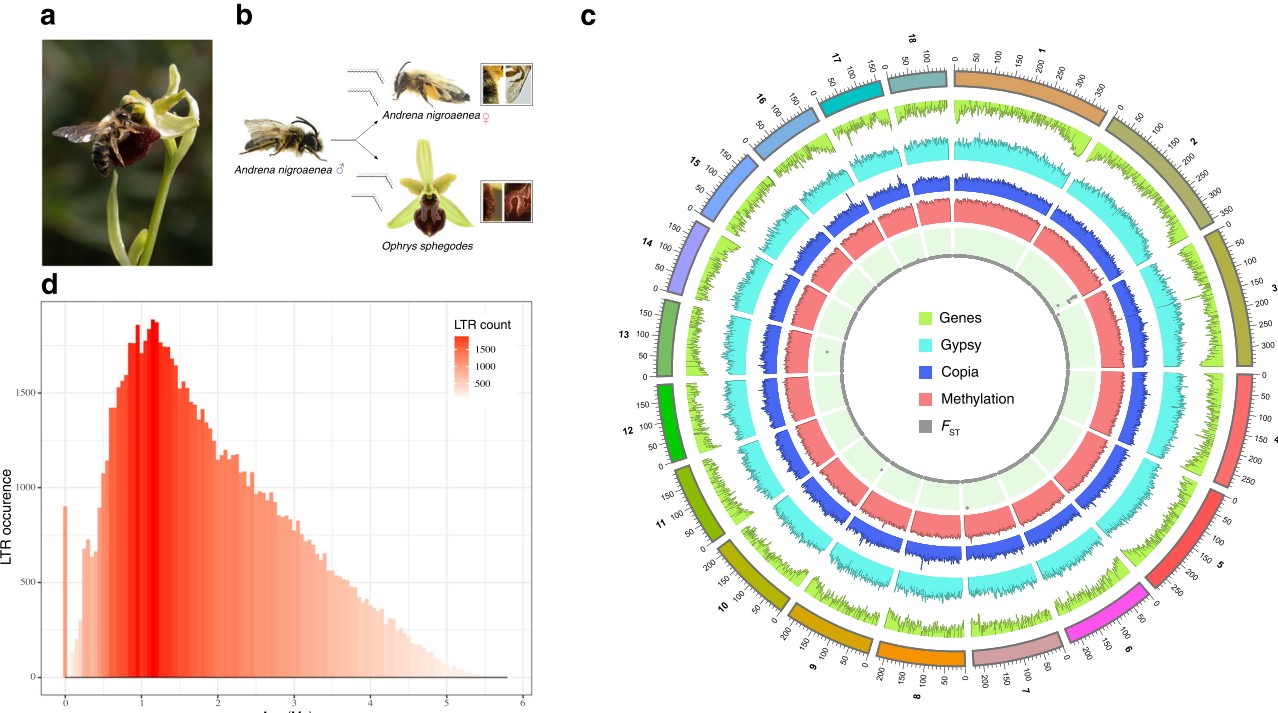

**Fig. 1 | Pollination and genome of *Ophrys sphegodes*. a** Pseudocopulation of an *Andrena nigroaenea* male with an *Ophrys sphegodes* flower. Photo courtesy of Noa Schwabe. **b** Schematic representation of sexual deception. An *Ophrys* orchid mimics the pollinator's female insect's hairs, wings and pheromones to lure the male into pseudocopulation. Photos courtesy of Noa Schwabe. **c** The 18 assembled pseudochromosomes of *O. sphegodes* are labelled from 1 to 18. From outside to inside: chromosomes; gene density in light green; *Gypsy* retroelements in turquoise; *Copia* retroelements in blue; cytosine methylome in light red; between-species differentiation (global $F_{ST}$) along chromosomes in grey. **d** Recent LTR insertions vs. their age, showing that LTR expansion reached its maximum at around 1.3 to 0.8 Ma ago.

(Benchmarking Universal Single-Copy Orthologs)[24]: 1453 out of 1612 (91.1%) conserved core land plant genes were found in our assembly, of which 1361 (84.9%) were complete (Supplementary Fig. 6a). We confidently annotated a total of 42,549 protein-coding genes, of which 90.0% had RNA-seq support or functional annotation (see Supplementary Tables 7, 8, Supplementary Note 1-2 and Supplementary Data 1–4 for genes of interest), which included information on Gene Ontology (GO) terms, protein domain information, putative pathways and enzyme function. Cytosine methylation was inferred from Nanopore sequencing data (Fig. 1c, Supplementary Figs. 7, 8).

## A burst in transposon activity preceded adaptive radiation

We manually characterised transposable elements (TEs) in the genome following a protocol[25] for TE classification that characterises TEs through a combination of homology-based prediction and manual inspection to find structural motifs and define sequence boundaries. To this end, we created a species-specific database of TEs for this orchid genome, containing a total of 436 sequences specific to *O. sphegodes*. Using this database, we identified a total of 4.05 Gb repetitive elements, occupying 78% of the *O. sphegodes* genome (Fig. 1c, Supplementary Table 9). Overall, *O. sphegodes* represents the largest orchid genome assembled to date and exhibits the highest abundance of long terminal repeat (LTR) elements (75% of the genome), more than *Vanilla planifolia* (10%)[26], *Apostasia shenzenica* (17%)[27], *Dendrobium catenatum* (40%)[27], *Phalaenopsis equestris* (44%)[27], *Gastrodia elata* (55%)[28], *Cymbidium sinense* (55%)[29], and similar to *Platanthera guangdongensis* and *P. zijinensis* (73% and 72%[30]; Supplementary Table 10). TE activity is known to influence genome size variation in eukaryotes[31], and LTR/*Gypsy* and LTR/*Copia* in particular have previously been shown to correlate with genome size expansion in orchids[32,33]. We conducted an analysis of LTR insertion age based on the idea that both LTR sequences of a TE are identical at the time of insertion, but will diverge over time as mutations accumulate. In *O. sphegodes*, analysis of recent LTR insertions showed that LTR activity had an initial increase at around 3 Ma ago, to reach its maximum at around 1.3 to 0.8 Ma (Fig. 1d). During this period, the Mediterranean Basin experienced climatic oscillations with glacial/interglacial periods[34,35]. It is conceivable that such environmental disturbances[36–38] might have led to bursts of TE proliferation in *O. sphegodes*, thus inflating its genome size. Interestingly, the peak of LTR element insertions precedes or overlaps with the radiation of the most species-rich *Ophrys* lineage, including the *O. sphegodes* species group, less than 1.0 Ma ago[19]. The ability to cope with genome size changes has allowed angiosperms to successfully diversify[39] and TEs have played an important role in enhancing angiosperm evolution[40] through their effects upon gene expression[37], as well as gene duplications and genomic rearrangements[41,42]. Since TEs often carry transcription factor binding sites, TE expansion can rewire existing metabolic networks and facilitate the evolution of new compounds (or mixtures), as was shown for the evolution of nicotine biosynthesis in tobacco[43]. Thus, TE bursts can contribute to the generation of intraspecific genetic and metabolic diversity[44]. Since changes in pollinator-attractive hydrocarbon compounds are suspected to be involved early in speciation in the *O. sphegodes* lineage[13], it is tempting to speculate that TE bursts may have provided this lineage with the genetic capacity to adapt to a new pollinator niche (i.e., *Andrena* bees), thereby facilitating the adaptive radiation of the *O. sphegodes* group.

## Genome evolution through chromosome fusions in the *Ophrys* lineage

To understand the evolutionary history of the *Ophrys* lineage, we constructed a phylogenomic tree and estimated divergence times across *O. sphegodes* and 20 other plant species with fully sequenced genomes, based on single-copy orthologues. *Ophrys* diverged from *Platanthera* (both from Orchidoideae subtribe Orchidinae), the most closely related orchid with a fully sequenced genome[30,45,46], approximately 22.42 Ma ago with a 95% confidence interval (CI) of 20.77–23.95 Ma, and Orchidoideae separated from other orchids around 54.49 Ma (CI 53.00–56.15 Ma; Fig. 2a, Supplementary Fig. 9). Our analysis further suggests that Orchidaceae separated from the common ancestor of Asparagales approximately 99.96 Ma ago (CI 98.66–105.66 Ma) and places monocot/eudicot divergence around 147.51 Ma (CI 139.87–154.45 Ma), in line with previous studies[47,48]. Our age estimates for orchids, while in line with results by Kim et al.[49], are younger when compared with previous orchid genome studies[27,30].

To track chromosome evolution of *O. sphegodes*, we compared it with the most closely related sequenced orchid genomes[30], focusing on the comparison with *Platanthera zijinensis* (Fig. 2b). The two genera differ in their chromosome numbers, with karyotype organisation of *Platanthera* ($n = 21$)[50] reflecting the ancestral and *Ophrys* ($n = 18$) the derived state[45,51]. Overall, most chromosomes maintained their structure between *Platanthera* and *Ophrys*, but some major rearrangements are apparent, particularly with regard to chromosome fusions. Chromosome (chr) 4 in *O. sphegodes* appears to be the product of a fusion between chr 7 and part of chr 4 in *Platanthera*, and *Ophrys* chr 10 derives from a fusion between *Platanthera* chr 15 and part of chr 14. Moreover, *Platanthera* chr 8 has no homologous chromosome in *Ophrys*, and significant parts of *Platanthera* chr 3, 4, 11, 14, 20 and 21 lack syntenic regions in *O. sphegodes* (Fig. 2b and Supplementary Fig. 10). Taken together, these findings are consistent with a reduction in chromosome number via fusions in the *Ophrys* lineage.

## Expanded gene families include genes involved in flower development

Sexual deception is not restricted to the Euro-Mediterranean genus *Ophrys*, but is a worldwide phenomenon. It has originated several times independently and there are many examples among Australian orchids, whereas only a few non-orchid cases, such as the South African daisy *Gorteria diffusa*[52], are known. Although specific pollinator interactions mediated by floral chemistry are a common theme in sexual deception[53,54], it remains unknown why this pollination strategy occurs predominantly among orchids and what allowed its repeated evolution in this family. To gain insights into the genomic basis of sexual deception in *Ophrys*, we first identified orthologous gene families using OrthoFinder[55]. We identified a total of 495,819 genes in 26,709 orthogroups among the 21 species, of which 3054 are shared among all species. A total of 1351 families containing 4537 genes were unique to *O. sphegodes*. We then identified expanded and contracted gene families using CAFE[56] (Fig. 2a). In *O. sphegodes*, 3712 gene families underwent an expansion, whereas 756 underwent a contraction. This is the highest level of gene family expansion reported among orchids to date, followed by *V. planifolia* (+3248) and *D. catenatum* (+1817). Of those gene families, 291 and 59 exhibited significant ($p$-value ≤ 0.01) expansions and contractions, respectively. Among the significantly expanded gene families were genes encoding transcription factors (TFs) involved in plant reproduction and flower development (but also other processes, e.g., stress responses), such as MADS-domain TFs (55 genes encoding type I and 6 genes of type II MADS; Supplementary Figs. 11, 12), MYB TFs (73 genes, of which 32 MYB-P; Supplementary Fig. 13), LATERAL ORGAN BOUNDARIES (LOB) domain TFs (10 genes), C2C2-GATA TFs (12 genes), WRKY TFs (11 genes) involved for instance in trichome development[57], SERINE CARBOXYPEPTIDASE-LIKE-1 (SCPL-1) proteins (15 genes) controlling anthocyanin acylation[58], TCP TFs (14 genes) regulating flavonoid biosynthesis and floral symmetry[59,60], and YABBY TFs (6 genes) involved in establishing adaxial-abaxial polarity[61]. The configuration of MADS-box genes putatively involved in perianth specification in *Ophrys* appears similar to other orchids (Supplementary Fig. 12), including the related *Orchis italica*[62], suggesting that the stark difference in their flowers likely results from the action of downstream genes. Secondly, disease

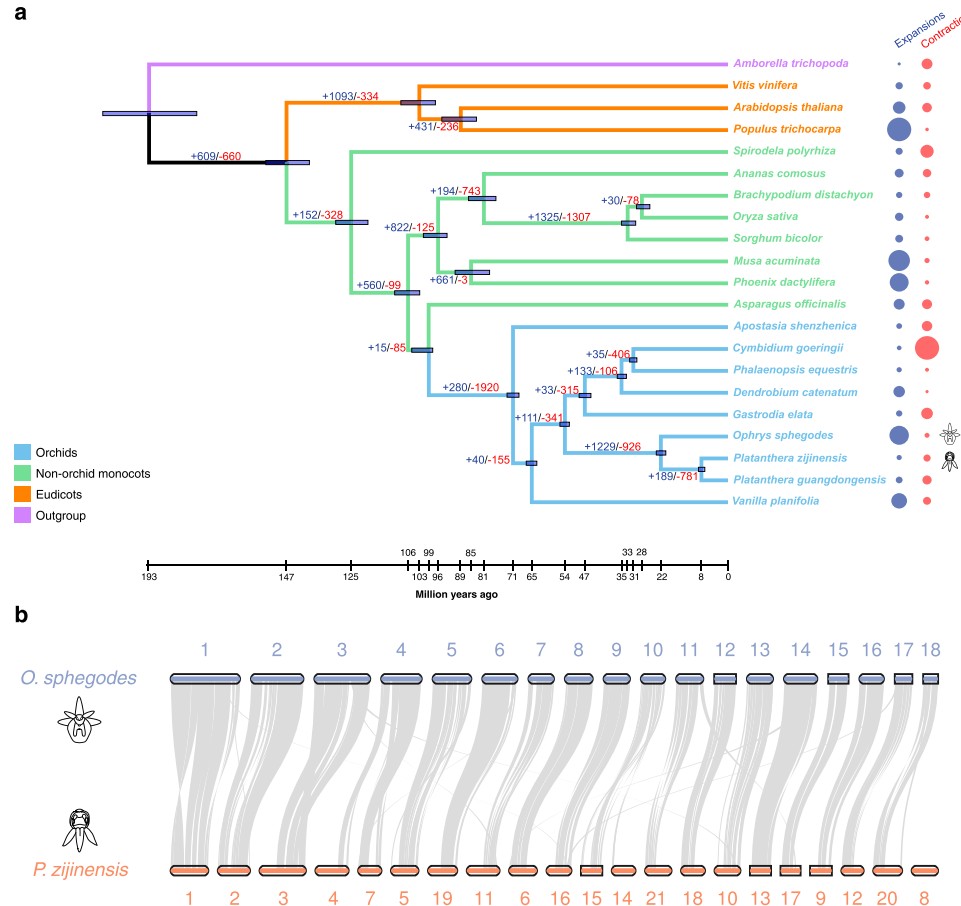

**Fig. 2 | Evolutionary relationships among orchid genomes. a** Phylogenomic tree showing estimated divergence time and evolution of gene families for 21 plant species, colour indicating different plant groups (see inset). Divergence times are indicated by light blue bars at the internodes; their ranges indicate the 95% confidence intervals (CI) of the divergence time. Numbers at the branches indicate the number of expanded (blue) and contracted (red) gene families in the lineages. Bubbles at the tips indicate expanded (blue) and contracted (red) gene families per species (see Supplementary Table 11 and Supplementary Fig. 9 for CI intervals and number of expanded/contracted gene families). **b** Genome comparison between *O. sphegodes* and *P. zijinensis*. Chromosome comparison shows a high degree of collinearity and some chromosome rearrangements (see also Supplementary Fig. 10).

resistance-related and stress response genes were found, such as NAC TFs (30 genes), glutathione S-transferases (28 genes) and plant PLEIOTROPIC DRUG RESISTANCE (PDR) proteins (10 genes). Finally, two previously identified putative candidate gene families for floral odour production and anthocyanin biosynthesis[63] showed significant expansion too, namely fatty acyl-CoA reductases (FARs; Fig. 3) and chalcone synthases (CHSs; Supplementary Fig. 14), also involved in defence response (29 and 14 genes, respectively).

**Plant adaptation to pollinators in *Ophrys* may involve the local duplication of candidate genes**

The key component for pollinator attraction in *Ophrys* species is the chemical mimicry of the pollinator female's sex pheromone, the composition of which has previously been characterised for *O. sphegodes s.l.*[10,64]. Alkene hydrocarbons are especially important, as different proportions of (Z)−12-alkenes, (Z)−9-alkenes and (Z)−7-alkenes are the major odour differences between *O. sphegodes* and the closely related *O. exaltata*, responsible for attracting different pollinators[65,66]. Thus, genes involved in hydrocarbon biosynthesis are likely important for pollinator attraction[63]. We annotated previously identified candidate genes[63] in the genome of *O. sphegodes* (Fig. 3, Supplementary Figs. 14 and 15, Supplementary Table 7, Supplementary Note 1 and Supplementary Data 1). Among these, structural annotation of stearoyl-ACP desaturases (SADs) showed that the key genes, *SAD1* and *SAD2* (*SAD2*-type), are duplicated in tandem in a single cluster on

chromosome 4 (283.17–283.30 Mb; containing four copies; Fig. 4d), whereas the house-keeping desaturase *SAD3*[67] resides in one copy in chromosome 5 (280.45–280.47 Mb) and *SAD4* (*SAD5*-type) is also present as a single copy (scaffold 75: 1.04–1.11 Mb; Supplementary Table 7 and Supplementary Data 1). For other *O. exaltata SAD5*-type genes, we identified four full-length copies in the *O. sphegodes* genome (at least two of them on the same scaffold 210), although none of these appeared to be a functional copy (Supplementary Table 7 and Supplementary Data 1). This is in line with previous findings that *O. sphegodes* only expresses functional *SAD2*-type alleles, while functional *SAD5*-type alleles are expressed in *O. exaltata*[66,67]. It is also consistent with gene expression patterns revealed by RNA-seq data (Supplementary Fig. 16). Phylogenetic analysis confirms the presence of separate *SAD* gene lineages corresponding to *SAD3, SAD2*-type (*SAD1/2/7/8*), and *SAD5*-type (*SAD4/5/6/9/10*, an incomplete *SAD11*) genes (Fig. 4a). Both *SAD2*- and *SAD5*-type lineages appear as single copies in *Platanthera*, suggesting they underwent recent gene duplication events. Of note, *SAD2* homologues were only present in orchid genomes from the subfamily Orchidoideae.

Like SADs, membrane-bound fatty acid desaturases (FADs)[68], which contribute to fatty acid desaturation and, potentially, alkene production[69], were also found duplicated in the *O. sphegodes* genome (Fig. 3). We found a phylogenetic lineage containing four *FAD* gene copies, of which at least three reside on chromosome 1 (368.58–368.63 Mb; Fig. 4e), a second lineage of two copies clustered

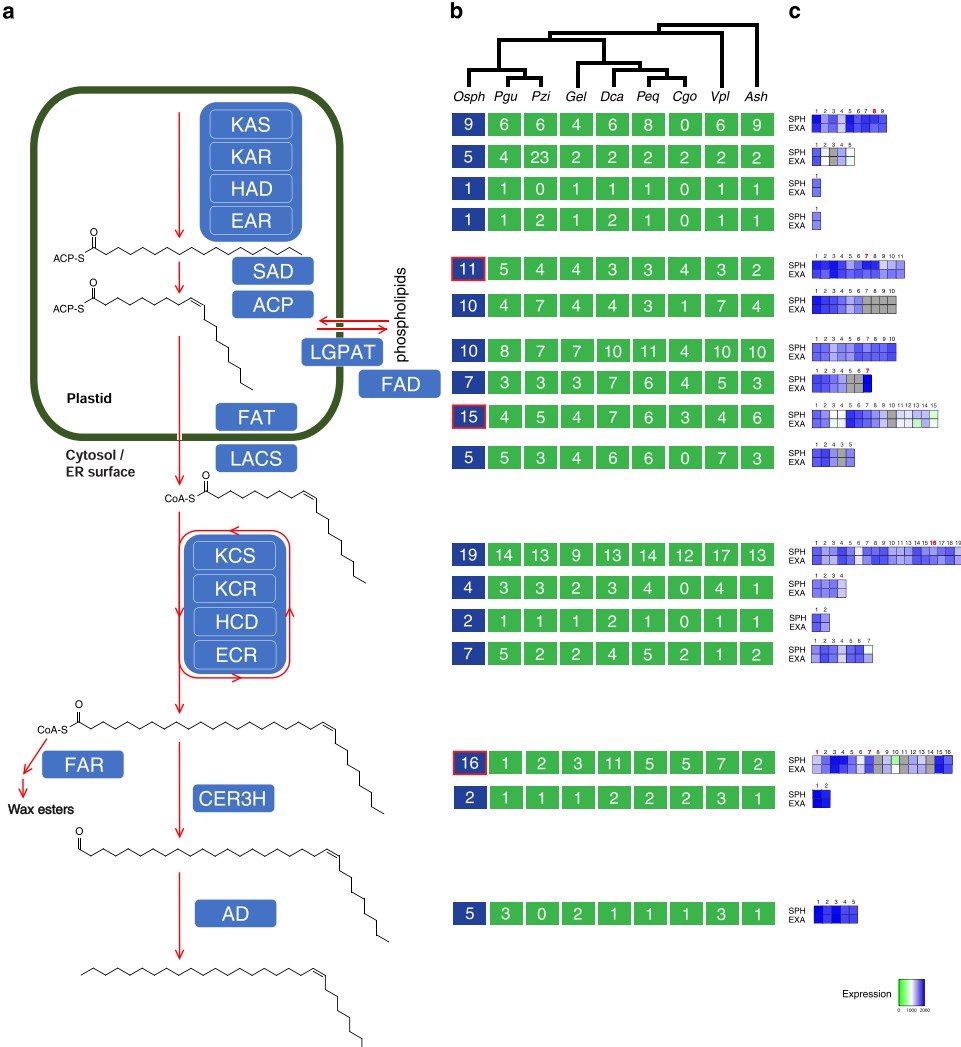

**Fig. 3 | Summary of the putative hydrocarbon biosynthesis pathway in *O. sphegodes*.** **a** Schematic representation of the core pathway showing the biosynthesis of a (*Z*)−9-alkene from fatty acyl precursors, depicting core proteins in blue boxes. **b** Gene copy numbers in the *Ophrys* (Osph, blue boxes) and other orchid genomes (green boxes), abbreviated by first letter of the genus and the first two letters of the species (as in Supplementary Table 11). Gene copy numbers were estimated by tallying the orchid members of orthogroups containing functionally annotated hydrocarbon biosynthetic genes. Gene families expanded in *Ophrys* are shown with a red outline. **c** Heatmaps of RNA-seq gene expression (green to blue; grey, not expressed) for *O. sphegodes* gene copies, showing expression for unpollinated mature flower labella of *O. sphegodes* (SPH) and *O. exaltata* (EXA). Numbers in red indicate significantly differentially expressed genes (FDR < 0.05).

on scaffold 33, while one gene, *FAD2*, was not duplicated. Both lineages showing gene duplications are present as single copies in the *Platanthera* genome (Fig. 4b).

Fatty acyl-CoA reductase (FAR) homologues show gene duplications in *Ophrys*. FARs likely catalyse the conversion of fatty acyl-CoA to primary alcohols and different FARs produce fatty alcohols with different acyl chain lengths[70]. Fifteen out of the 16 *FAR* homologues found in the *O. sphegodes* genome form a phylogenetic Orchidoideae gene clade together with a single *Platanthera* sequence. Of these, 14 genes are distributed over only three scaffolds (Fig. 4c; chr 5: 1; scaffold 279: 4; scaffold 133: 4; scaffold 578: 6). Six of these clustered *FAR* genes reside in one region of 109 kb on scaffold 578 (Fig. 4f). The close vicinity of these copies in a unique clade shows a recent duplication event in the *O. sphegodes* lineage (Fig. 4c). Gene duplication plays a crucial role in shaping the evolutionary landscape of genomes, as they provide the main raw material for the evolution of new genes[71]. Single or tandem gene duplications are also involved in the origin of many plant genes[72]. Often, retention of duplicate genes occurs non-randomly, as changes in the concentration of gene products can have a selective advantage for the organism[73]. These genes may provide

*Ophrys* with an opportunity to respond to selection by pollinators, e.g., through positive dosage effects or neofunctionalisation and pseudogenisation of the less effective variants[74].

## Population genomic analyses reveal putative barrier loci under pollinator-driven selection

Closely related *Ophrys* species provide plausible examples of pollinator driven-speciation[13,18]. We therefore investigated the genetic differentiation between *O. sphegodes* and three other closely related, sympatric co-flowering species, *O. exaltata*, *O. garganica* and *O. incubacea* in Gargano, Southern Italy[12,13]. These four species are pollinated by sexual deception of different solitary bee males: *Andrena nigroaenea*, *Colletes cunicularius*, *A. pilipes* and *A. morio*, respectively[75]. These *Ophrys* species show variation in floral traits, ranging from labellum coloration (markedly blacker in *O. garganica* and *O. incubacea*[13]), to different floral odours mimicking their pollinators' sex pheromones, which make a major contribution to reproductive isolation[12,13,66,76]. We used Genotyping-By-Sequencing (GBS) data[13] to investigate the genetic differentiation between species and scanned the genome for signatures of selection. Chord between-population genetic distance[77]

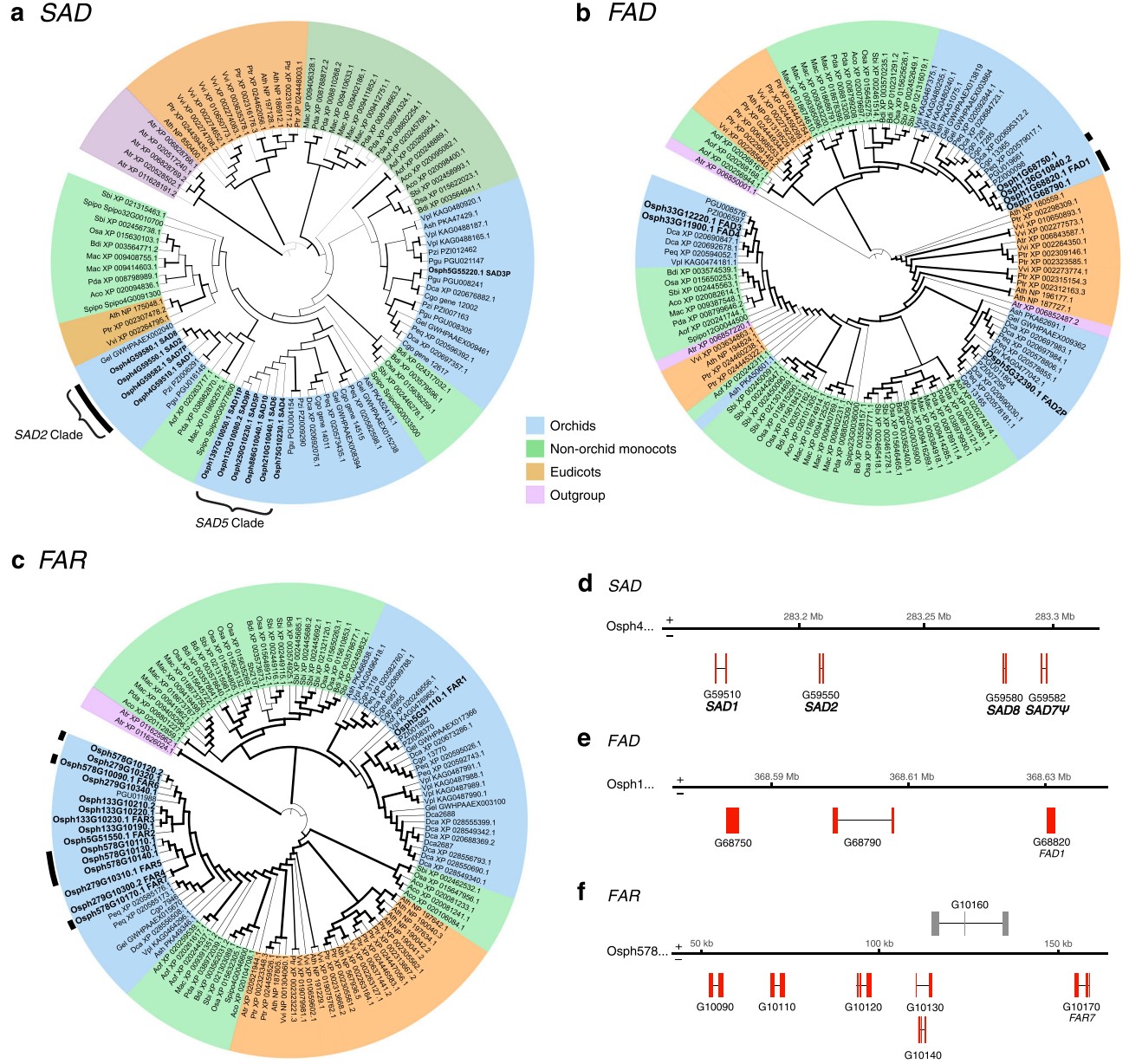

**Fig. 4 | Phylogenetic analyses of three gene families showing gene duplications.**
**a** *SAD* gene family tree. **b** *FAD* gene family tree. **c** *FAR* gene family tree. Colour shading for taxa as in Fig. 2a. *O. sphegodes* genes are shown in bold, branch thickness indicating bootstrap support. The first three or four letters of each gene sequence indicate species (as in Supplementary Table 11), where Osph indicates *O. sphegodes*; Pgu and Pzi indicate *Platanthera guangdongensis* and *P. zijinensis*, respectively. Black bars highlight clustered genes shown in subsequent Figure panels. **d** *SAD2* gene cluster on chr 4, containing *SAD1/2/8/7ψ* (gene details in Supplementary Table 7 and Supplementary Data 1). **e** Details of a cluster on chr 1 containing 3 *FAD* homologues. **f** Details of scaffold 578 with 6 *FAR* gene homologues. Gene models of interest are drawn in red.

was calculated per 1 Mb window to identify the most similar/dissimilar species at a given chromosomal region in the *O. sphegodes* genome. This analysis revealed that segregating polymorphisms between species are distributed across the genome, and that overall genetic (dis)similarity between *O. sphegodes* and the three other species is roughly equal (Fig. 5a, Supplementary Fig. 17), as may be expected in a species radiation. Yet, cumulatively, the four species were clearly separable in a principal coordinate analysis (PCoA; Fig. 5b; Supplementary Fig. 18 for individual chromosomes). These findings are in line with previous population analyses, suggesting that many polymorphisms in the genome are shared among all species[13], whilst few barrier loci may separate them. Furthermore, global $F_{ST}$ outlier analysis (FDR < 0.01) of a GBS dataset from 126 individuals[13] revealed a highly differentiated region in chr 2 (333–352 Mb; Figs. 1c and 4c; Supplementary Fig. 19).

Interestingly, in contrast to the genome-wide pattern, especially *O. sphegodes* and *O. exaltata* are clearly separated in the highly differentiated ~20 Mb region on chr 2, as seen by PCoA of this region (Fig. 5d). In an independent $F_{ST}$ analysis based on RNA-seq data, this genomic region was confirmed as being differentiated between these species, although this dataset offers denser sampling of the genome and yielded a somewhat larger interval (327 – 358 Mb, 31 Mb in length; 0.6% of the genome; Supplementary Fig. 19 and Supplementary Data 5). Whether this pattern of differentiation is due to divergent selection and suppression of effective recombination via, e.g., divergence hitchhiking or an inversion, is currently unknown. Since the differentiated region of chr 2 did not contain any a priori candidate genes for hydrocarbon or pigment biosynthesis (Supplementary Table 7 and Supplementary Data 1, 2), we performed a GO

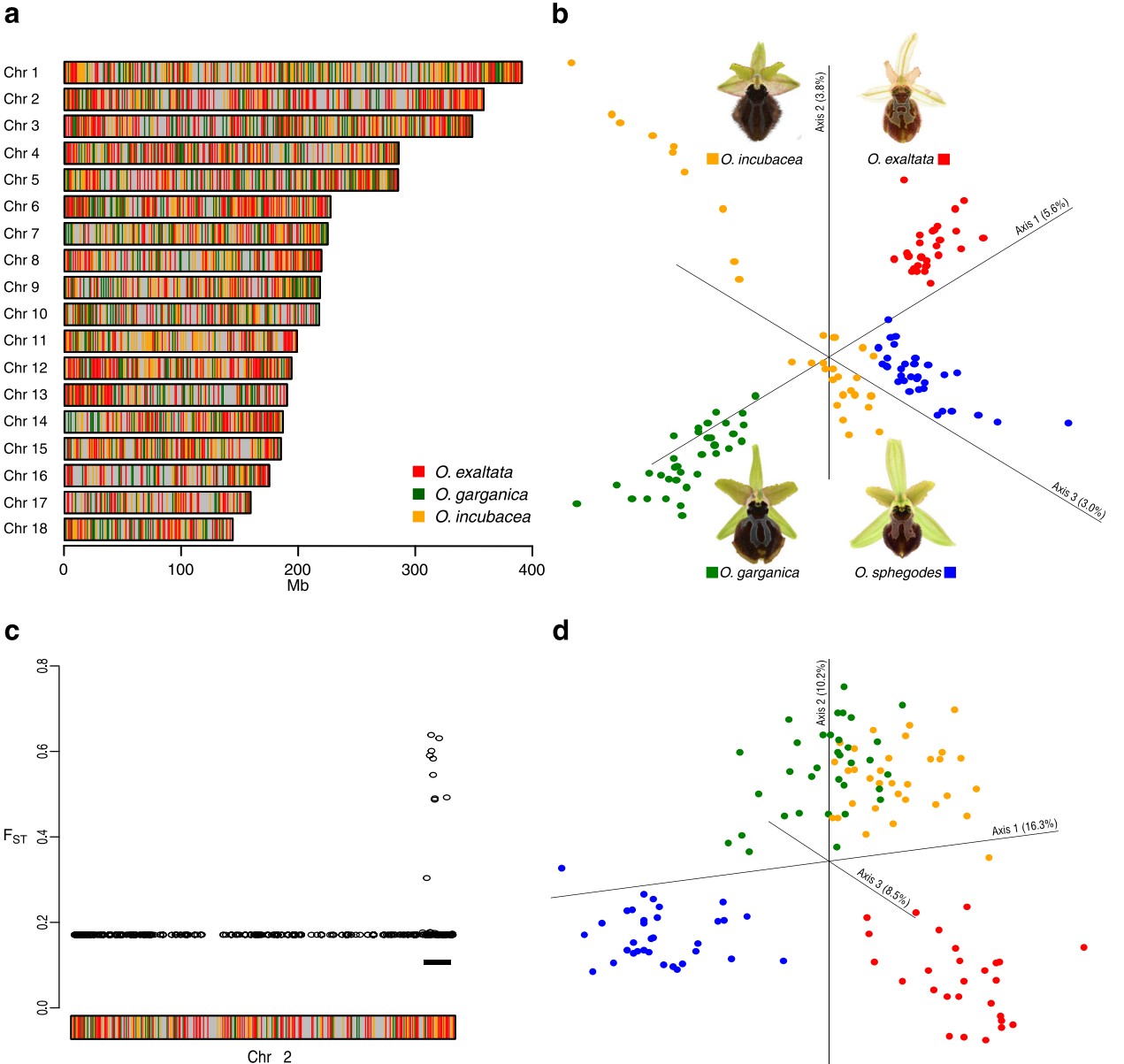

**Fig. 5 | GBS-based population genetic analysis between four sympatric *Ophrys* species. a** Genetic distance among the different species of the *O. sphegodes s.l.* group plotted along the genome, colours showing the most genetically dissimilar species for a given 1 Mb window (see Supplementary Fig. 17 for genetic similarity). **b** Genome-wide PCoA plot of pairwise distances between plant individuals revealed genetic separation between the four species. **c** Global $F_{ST}$ among the 4 species was elevated in one region on chr 2 (327–346 Mb, shown as black bar). **d** PCoA of this region of chr 2 shows increased separation between *O. sphegodes* and *O. exaltata*. Colours show the most dissimilar species in panels **a** and **c** (grey: no data), and species identity in panels **b** and **d**, where *O. sphegodes* is shown in blue; *O. exaltata*, *O. garganica* and *O. incubacea* are shown in red, green and yellow, respectively.

enrichment analysis (Supplementary Data 6). This revealed term GO:1903415 ('flavonoid transport from endoplasmic reticulum to plant-type vacuole') as the most significantly enriched 'biological process' category. Three genes (*Osph2G68830*, *Osph2G68850* and *Osph2G68960*) with this term are homologues of *Arabidopsis TT9* (*AT3G28430*), a peripheral membrane protein necessary for vacuole development and the accumulation of anthocyanins in the vacuole[78]. Additionally, we screened genes in this region for (i) elevated $F_{ST}$ between *O. exaltata* and *O. sphegodes*, (ii) annotated transcription factors, (iii) genes with excess amino acid change (i.e., excess non-synonymous nucleotide diversity), and (iv) differential floral labellum gene expression between the two species (Supplementary Fig. 19). Only 7 genes were found in more than one screen (Supplementary Fig. 19d, Supplementary Data 7), three of which had useful annotation information that revealed one AP2 and one B3-ARF TF (*Osph2G66470*

and *Osph2G63210*, respectively) as well as one ubiquitin conjugating enzyme (*Osph2G63100*). While it remains to be established whether any of these genes are directly linked to differences between the two *Ophrys* species, currently it seems more likely that regulatory rather than biosynthetic genes contribute to this genomic region of differentiation. Overall, using the metaphor of genomic islands of speciation[79], these results suggest that adaptation involving individual loci (or small genomic segments) rather than entire genomes, characterises these reproductively isolated sympatric *Ophrys* species. The overall high level of allele sharing and the genomic mosaic pattern of species relationships (without long contiguous stretches of similar relationships) cannot easily be explained by gene flow after a secondary contact of species that are separated by strong floral isolation[12]. An alternative explanation could be that the high level of segregating polymorphisms is due to shared ancestry and large effective

population sizes, suggesting that the species are in an early stage of genomic divergence[11,12,65,79]. Our population genomic analyses identified candidate regions potentially under pollinator-driven selection, thus calling for future research into the roles and functions of the genes in these regions.

The chromosome-level genome sequence of *O. sphegodes*, a plant with high pollinator specificity, provides important insights into the evolution of plant adaptation to pollinators and its role in species diversification. It seems that, ecologically, the recent adaptive radiation of this group was fuelled by the availability of pollinator niches and, genetically, a burst in TE activity and rampant gene duplication provided the genetic raw material for pollinator-mediated selection to act upon. These mechanisms may also provide a blueprint for other angiosperm radiations, as floral trait-mediated reproductive isolation may often be underlain by a simple genetic architecture[2]. More broadly, this case study supports the idea that the generation of genomic diversity, as genomic potential in the form of genome duplication, hybridisation or TE activity, often precedes adaptive radiation.

## Methods

### Sample preparation

The single *O. sphegodes* individual for the genome assembly (accession SPH_8) was selected among several samples previously collected in Capoiale, Gargano area, Southern Italy[12], and grown in a pot at the Dept. of Biology, University of Naples Federico II, under natural light at ambient temperature. Fresh young leaves were harvested, snap-frozen, and high-molecular-weight genomic DNA was isolated with an SDS lysis buffer supplemented with β-mercaptoethanol, followed by purification via phenol/chloroform extraction and carboxylated magnetic beads[80].

### Genome size estimation

Nuclear DNA content and thus genome size was estimated via flow cytometry. Since genome size estimation via flow cytometry from orchid leaves is challenging due to DNA endoreduplication[81], we used pollinia (carrying haploid pollen) for the analysis. We followed a previously published protocol[82] with slight modifications. Briefly, a pair of pollinia was crushed in a 1.5 ml tube with Otto I buffer using a clean pestle, transferred to a Petri dish, and co-chopped with $2 \times 2$ cm tissue of a reference leaf (*Solanum lycopersicum* cultivar 'Stupicke polni tyckove rane'; tomato 1C = 0.98 pg, as measured in ref. 83). The suspension was filtered, mixed with Otto II buffer, and stained with propidium iodide in the dark at 4 °C for 1 hour. At least 10,000 nuclei were analysed on a Cytoflex S (Beckman Coulter) flow cytometer. The average fluorescence value (Mean PerCP-A, see Supplementary Fig. 1) of reference 2C nuclei was 4,207,840; the average fluorescence for *O. sphegodes* 1C nuclei was 10,506,736 (Supplementary Table 1). Thus, following a published formula[84], and converting picograms of DNA to number of nucleotide pairs[85], we estimated a haploid genome size of 4.83 Gb (average of three measurements, see Supplementary Table 1).

### Nanopore library construction, sequencing, and genome assembly

DNA isolated from accession SPH_8 was used to prepare two Illumina and eight ONT libraries (Supplementary Table 2), following the general guidelines provided by Oxford Nanopore Technologies® for the 1D Genomic DNA by Ligation (SQK-LSQ109) protocol, with modifications proposed by New England Biolabs® (NEB)[80]. Six ONT libraries were sequenced on a PromethION PTC0031 platform (Oxford Nanopore Technologies), and two on MinION Mk1B. All sequencing was performed at Agroscope Wädenswil, Switzerland. We used Miniasm v0.3 and Minimap2 v2.17[23] to assemble the initial contigs. Pilon v1.23[86] and Racon v1.4.3[87] were used to correct indels and mis-assemblies with Illumina reads (Supplementary Table 3 and Supplementary Method 1). To deal with the heterozygosity of the *O. sphegodes* genome, we

used Redundans v0.11[88] to remove under-collapsed contigs and obtain a final assembled genome size of 5.2 Gb (see Supplementary Method 1). Raw Nanopore sequencing data were subsequently re-basecalled with Guppy v5.0.11, and Nanopolish v0.13.3 was used to detect 5-methylcytosine (see Supplementary Method 2).

### Hi-C library preparation and assembly scaffolding

A total of 0.5 g leaf tissue were fixed in nuclei isolation buffer and 36% formaldehyde, followed by cell lysis, chromatin digestion with the enzyme *Hin*dIII-HF (New England Biolabs, Ipswich, MA, USA, R3104L), re-ligation, DNA re-extraction and library preparation[89,90] as detailed in Supplementary Method 1. Four Hi-C libraries were sequenced on an Illumina NovaSeq 6000 SP FlowCell. To assemble the chromosomes, raw Hi-C data was mapped against the genome assembly using the ArimaGenomics mapping script (https://github.com/ArimaGenomics/mapping_pipeline/blob/master/01_mapping_arima.sh). Subsequently, Salsa v2.3[91,92] was used to build scaffolds (total scaffolds 2511, N50 = 4.8 Mb); Juicebox Assembly Tools (JBAT)[93] and the 3D-DNA pipeline v-180114[94] were used to order scaffolds to chromosomes.

### Gene and non-coding RNA annotation

Gene annotation was carried out via an integrative approach that included de novo gene prediction, homology-based prediction and transcriptome-based prediction. AUGUSTUS v3.4.0 was used as part of the BRAKER2 v2.1.6 pipeline[95] and trained with RNA-seq data[20] from floral tissues to predict coding regions in the repeat-masked genome. ProtHint v2.6.0 (https://github.com/gatech-genemark/ProtHint) was used to generate protein hints using the protein database liliopsida_odb10-v.2020-09-10 from OrthoDB10, to score intron intervals, start and stop codons from ultra-conserved proteins of the monocot lineage. This extrinsic evidence was given to GeneMark-EX v4.64[96] to self-train, and improve prediction accuracy of AUGUSTUS. Proteomes from *P. equestris*[27], *C. goeringii*[97], *O. sativa* ssp. *japonica* cultivar Nipponbare IRGSP-1.0[98], and *A. officinalis* Aspof.V1[99] were downloaded and used as input for GeMoMa v1.8[100] for homology-based prediction. The transcriptome of *O. sphegodes*[20] was mapped against the genome using PASA v2.5.1 (Program to Assemble Spliced Alignment; https://github.com/PASApipeline/PASApipeline/blob/master/docs/index.asciidoc). Valid transcript alignments were clustered based on genome mapping location and assembled into gene structures. PASA assemblies were incorporated into gene predictions to correct exon boundaries, add UTRs, and update gene structures. AHRD v3.3.3 (https://github.com/groupschoof/AHRD) was used to filter genes based on SwissProt, trEMBL and TAIR10 databases, and candidate genes for pollinator attraction from previous studies were manually included. Functional annotation was carried out using TRAPID v2.0[101] for Gene Ontology (GO) terms, InterPro-Scan, KEGG, and iTAK Classifier[102] for transcription factors and protein kinases. Genes for tRNA and rRNA were identified using tRNAscan-SE v2.0.9[103] and Barrmap v0.9 (https://github.com/tseemann/barrnap; see Supplementary Method 3 for details).

### Repetitive element annotation

We used a published approach[25] to classify and manually annotate repetitive elements in the genome. Putative TEs identified with RepeatModeler v2.0.1 were blasted against the TRansposable Elements Platform (TREP) database (v.2019, https://trep-db.uzh.ch/). Sequences with strong hits to a known family were selected (plus their flanking regions) and blasted against the *O. sphegodes* genome to identify other copies belonging to the same TE family. Multiple sequence alignments between those clusters of sequences were performed to create consensus TE sequences, and to search for structural motifs that could help classify them. Sequences with no strong hits to a known family were blasted against PTREP (Protein TRansposable Elements Platform) or Dfam (https://dfam.org) or examined for the presence of structural

motifs to identify TEs in the genome. All TE sequences were classified according to a three-letter code[25].

## Analysis of LTR divergence and insertion time

LTR_harvest v2.0.0[104] was used to identify LTRs with complete structure (-xdrop 37 -motif tgca -motifmis 1 -minlenltr 100 -maxlenltr 3000 -mintsd 2), expected to be detectable for young LTR copies. The insertion time was estimated based on the sequence divergence between two LTRs of the same type. 5' and 3' LTR of each full-length paralogue were aligned using Mafft[105] and divergence between them was estimated using the Kimura two-parameter model using the distmat program implemented in the EMBOSS package v6.6.0 (https://emboss.sourceforge.net/). LTRs insertion age was estimated using a substitution rate of $1.3 \times 10^{-8}$ substitutions per site per year[106].

## Phylogenomic analysis and estimation of divergence time

The protein-coding regions of 21 plant species with a sequenced genome (Supplementary Tables 11 and 12) were used for phylogenomic analysis. Orthologue groups were identified using OrthoFinder v2.5.4[55], using Mafft for multisequence alignment and fastTree[107] for building the trees. Among 526,955 genes from the 21 species, 495,819 (92.1%) were assigned into 26,709 orthologue groups. The concatenated supermatrix of 34 single-copy nuclear genes was used to infer phylogenetic relationships using the GTR-GAMMA model with 1000 bootstrap replicates in RAxML v8.2.11[108]. The same supermatrix and corresponding concatenated ML tree were used for dating analysis. We estimated divergence times using the penalised likelihood method implemented in TreePL v1.0[109]. We considered 83.78 Ma and 114.92 Ma as the minimum- and maximum-age calibrations of the stem age of orchids[49]. We used an optimal smoothing parameter determined by the "random subsample and replicate" cross-validation method to accommodate rate heterogeneity. A total of 1000 bootstrap replicates with the topology fixed to the concatenated ML tree and branch lengths allowed to vary were also generated using RAxML for calculating the confidence intervals of age estimates. Results from the dating of the bootstrapped trees were then summarised and visualised on the concatenated ML tree using TreeAnnotator (part of the BEAST2 v2.7.0 package[110]).

We used CAFE v5.0.0[56] to identify gene family contractions and expansions, using the reconstructed species tree. Only gene families were kept in which among-species difference in size was less than 100. First, nucleotide sequences of the previously identified single-copy orthologues were translated into amino acids with TranslatorX v1.1 within the MitoPhAST-master v3.0 package[111], and then given as input to MUSCLE v5.1.0[112] for multiple sequence alignment (MSA). The resulting MSAs were trimmed with trimAL v1.2[113] (parameter -automated1 to select optimal cut-off based on alignment similarity) to remove poorly aligned regions or spurious sequences, and the best-fit model for each tree was computed with ModelTest-NG v0.1.7[114] (parameters -f ef -h f -s 3 -tr phyml). The final trees were built with phyml v3.3.20220408 (parameters -d nt -c 4 -m AIC -f modelfreq -o modelparam) (https://github.com/stephaneguindon/phyml) using the parameters from ModelTest-NG and 1001 bootstrap replicates.

## Comparative genomic analyses

Synteny analyses between *O. sphegodes* and *P. guangdongensis* and *P. zijinensis* were performed using the MCScan[115] functions of the JCVI utility libraries v1.2.9 (https://github.com/tanghaibao/jcvi). First, *jcvi.compara.catalog ortholog* (--min_size 5 -dist 35) used LAST v2.34[116] to identify syntenic blocks between the *O. sphegodes* genome and another genome (see also Supplementary Table 13). Second, *jcvi.compara.synteny screen* (--minspan 30 -simple) was used to create simplified versions of the anchor files containing the syntenic blocks. Finally, linear synteny plots were made with *jcvi.graphics.karyotype* and dot-plots with *jcvi.graphics.dotplot*.

## Population genetic analyses

Previously published GBS data from 126 individuals[13] and RNA-seq data from floral tissues of 32 accessions[20] were used to analyse the genetic variation between *Ophrys* species. Both datasets were mapped against the genome using BWA-MEM2 v2.2.1[117] with default parameters, and variants (SNPs and small InDels; Supplementary Tables 14 and 15) were identified using freebayes v1.0.2 (https://github.com/freebayes/freebayes) with minimum depth 2 (-C 2). Samtools v1.10[118] was used to remove duplicated reads prior to variant calling. Raw variants (1,680,249, of which 1,621,433 biallelic GBS markers) were filtered for depth and frequency, and pairwise genetic distances (comparison matrix in Supplementary Table 16) calculated between individuals using SPA v0.1 (https://peb.uni-hohenheim.de/spa) as detailed in Supplementary Method 4. Biallelic SNPs were used with BayeScan v2.1[119] to identify candidate loci under selection in a "global" manner, i.e., treating each species as a subpopulation, and in a "pairwise" fashion, comparing two species at a time. Chord distance[77,120] was calculated and PCoA was carried out genome-wide and per chromosome. A detailed description of population genetic methods can be found in Supplementary Method 4 and, for analyses of chromosome 2, in Supplementary Method 5.

## Reporting summary

Further information on research design is available in the Nature Portfolio Reporting Summary linked to this article.

## Data availability

The genome assembly and raw sequencing data generated for this study, including ONT data, PacBio data, Illumina WGS and HiC data, were submitted to NCBI under BioProject number PRJNA994461. The Whole Genome Shotgun project was deposited at DDBJ/ENA/GenBank under the accession JBANGT000000000. The version described in this paper is version JBANGT010000000. The RNA-seq data used for genome annotation and expression analysis can be found on NCBI under accession PRJNA574279[20]. The GBS data used for population genetic analysis can be found in the NCBI accession PRJNA257331[13]. Annotation data, including protein-coding gene annotation, transposable element database and annotation, non-coding RNA, as well as the alternative haplotig fasta file can be found on figshare [https://doi.org/10.6084/m9.figshare.25398166]. Transposable element sequences were also deposited in the TREP database [https://trep-db.uzh.ch/]. Source data are provided with this paper.

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

## Acknowledgements

We are grateful to Martin Mascher and Chang Liu for fruitful discussions about the project, to Yangzi Wang and Jhih-Sheng Liu for assistance with analyses, and to Noa Schwabe for kindly providing pictures of *Ophrys* and *Andrena*. This work was supported by the University of Zurich, a Ph.D. project funded by the University Research Priority Program "Evolution in Action: from Genomes to Ecosystems" (to U.G. and P.M.S.), the Swiss National Science Foundation (SNSF; grant 31003A_155943 to P.M.S. and grant IZLRZ3_163885 to U.G.), the German Research Foundation (DFG; project 446145319 to P.M.S.) and the 2022 PRIN Program (to S.C.). This work is set within the framework of the "Laboratoires d'Excellences (LABEX)" TULIP (ANR-10-LABX-41) and of the "École Universitaire de Recherche (EUR)" TULIP-GS (ANR-18-EURE-0019). This work made use of infrastructure services provided by the Science IT team of the University of Zurich (www.s3it.uzh.ch) and was supported via bwHPC by the High Performance and Cloud Computing Group at the Zentrum für Datenverarbeitung of the University of Tübingen, state of Baden-Württemberg, and the DFG (grant INST 37/935-1 FUGG).

## Author contributions

A.R., P.M.S. and U.G. conceived and designed this study. A.R., P.M.S. and S.C. provided *Ophrys* samples. A.R. performed genome size estimation, gDNA extraction and the Hi-C experiment. A.R. and D.S. conducted the initial genome assembly. A.R. improved the assembly and performed annotation analysis. D.F. performed Nanopore library preparation and sequencing. A.R. and M.A. performed the JBAT visualisation to final chromosome scaffold. A.R., T.R.G., L.G. and T.W. conducted the transposable element annotation. M.E.B. conducted the LTR divergence time analysis. H.F.O. performed the methylation analysis. S.X., Q.T., R.E.O. and S.E.G.M. carried out the phylogenetic and gene family analyses. G.P. conducted the synteny analysis. A.R. and P.M.S. performed the population genomic analyses. M.A. performed differential expression and GO analysis. A.R. and M.A. performed figure curation. A.R. wrote the original draft of the manuscript with inputs from P.M.S., S.C., S.X. and U.G. All authors discussed and approved the final manuscript. J.E.F., J.B., S.C., O.P. and U.G. provided critical resources for this study. P.M.S. and U.G. supervised the project. P.M.S., U.G., S.C. and J.E.F. acquired funding.

## Funding

## Competing interests

The authors declare no competing interests.
