## [Peer Review File · Nature Communications]

Genome of the early spider-orchid *Ophrys sphegodes* provides insights into sexual deception and pollinator adaptationReviewers' Comments:

Reviewer #1:

Remarks to the Author:

Russo et al. reported the chromosome-scale genome sequence of *Ophrys sphegodes*, providing evidence for chromosomal rearrangements in the *Ophrys* lineage. They also identified a transposable element expansion event in the *O. sphegodes* group and highlighted the role of gene duplication in contributing to the evolution of chemical mimicry. In addition, the authors proposed a candidate genomic region associated with pollinator-mediated evolution. I have read the preprint of the manuscript in October and found it intriguing. However, some major revisions are necessary for it to be suitable for publication in Nature Communications.

1. This work conducted whole-genome sequencing to unravel the mechanisms underlying sexual deception in *Ophrys sphegodes* and its adaptation to pollinators that fosters reproductive isolation and species differentiation. The manuscript centers around genomic sequencing but falls short of fully utilizing genomic datasets to address significant questions: (1) Despite the challenge posed by the highly-modified nature of sexually deceptive flowers, it's suggested to perform genome evolution and comparative analyses to investigate the origin of sexual deception pollination in *Ophrys*. (2) Metabolic data should be combined with whole-genome data to reveal the sex phytohormone pathway in *Orchrys*. (3) Leveraging the whole-genome data in conjunction with population genetic structure to elucidate the functional differentiation mechanisms of sexual deception in *Ophrys* species.

2. Most of the discussion seems speculative without providing robust analysis and evidence: (1) In the section titled "A burst in transposon activity preceded adaptive radiation", the authors suggested that TE insertions led to the adaptation of new pollinators, relying solely on literature evidence without presenting results supported by original analysis. It is also unclear how the TE burst enabled *Ophrys* to adapt to a new pollinator niche and facilitate adaptive radiation, which is an important new insight of the manuscript. (2) For the discussion on "Expanded gene families may be involved in flower development", the evidence is based solely on a paper published in 2013. In addition, only in the supplementary 10 showing the phylogeny of MADS-box and MYB genes. However, I do not understand each of the subfamily represent what kind of MADS-box gene as well as MYB genes. The authors also indicated LOB domain TFs, C2C2-GATA, WRKY, TCP, YABBY, NAC TFs might involve in development. However, actually many of these TFs not only involved in the development but many other aspects of biological processes, such as biotic and abiotic stress responses. It is recommended to conduct analyses of functional genes based on genomic evidence and add analysis on flower morphogenesis and pigmentation, which is key to provide evidence for how flower morphology and color simulate the perception of pollinating insects. (3) Again, in the section titled "Plant adaptation to pollinators in *Ophrys* is associated with local duplication of candidate genes", additional transcriptome evidence is needed to support the duplication genes.

(4) In "Population genomics analyses reveal putative barrier loci under pollinator-driven selection", there seems to be inconsistency between the discussion of chr2 and the results shown in Fig3 (Chr4). Please further analyze the highly-differentiated c. 20Mb region on Chr2, including genome annotation and functional enrichment analysis. It is important to show whether key genes related to sexual deception are located in this region and, if not, provide an explanation for how this region contributes to pollination isolation.

3. Quality and assessment of sequencing data. It is necessary to further confirm the following genome assembly and annotation results: The assembled contig N50 value (754kb) appears to be relatively low compared to current sequencing technologies for genomes. Can this be improved? The BUSCO values are acceptable, while this may be attributed to the inherent characteristics of Nanopore sequencing. Nanopore sequencing may also result in elevated annotated protein-coding genes compared to other sequenced orchids. This overrepresentation includes an excess of MADS-box genes and transcription factors (TFs), as well as the duplicated genes, which need to be further confirmed.

Minor questions:

1. Lines 26-30 are verbose; a concise statement of scientific significance should conclude line 37.
2. The discussion of pollination-mediated evolution needs a more direct link to the title. The cited literatures in lines 53-55 are inconsistent with the genomic heterozygosity (1.28%).
3. Lines 97-98, the LTR of *Vanilla planifolia* is 10%?
4. Lines 104-108, hybridization facilitated by climate events seems speculative without supporting evidence, and lacks a logical connection with reproductive isolation.
5. Lines 119-128, the differentiation time for Orchidaceae is inconsistent with that in the published orchid genome papers. Please provide an explanation.

Reviewer #2:

Remarks to the Author:

This exciting and highly readable manuscript reports the 5.2 GB chromosome-scale genome sequence of the sexually deceptive orchid, *Ophrys sphegodes*. While genome sequencing, assembly and annotation is increasingly feasible, an effective team of multi-disciplinary specialists is still required. It is clearly evident that such an expert team has been involved in this project.

Collectively this outstanding team have achieved several key firsts. For example, from a genome perspective, this team has successfully tackled the largest orchid genome to be sequenced to-date. Further, while it is not surprising that such a large orchid genome will be characterised by the highest abundance of long terminal repeat elements (relative to other sequenced orchid genomes), this study has taken on the challenge of building the first species level database of orchid transposable elements. Such a database promises to be a powerful resource for many future studies. The study also anchored the new orchid genome with a phylogenomic analysis, divergence time estimation and analysis of genome size expansions and contractions across a total of 21 orchid and outgroup genomes. Finally, this is also the first orchid genome to be sequenced, where the species is obligately dependent on a single pollinator species, in the highly specialised pollination strategy of sexual deception, offering an exemplar case study where pollinator-driven evolution is likely to leave a strong genomic signature.

The strategically chosen study species, *Ophrys sphegodes*, has been the subject of prior in-depth ecological, evolutionary, chemical and molecular studies, while the genus at large is one of the most intensively studied of all wild plant genera. Building on this exceptionally rich background knowledge, this study has gone beyond other genome papers to draw strong connections and links between this background knowledge, and new findings from the genome of a wild species. As one example, the study included detailed investigations of three gene families predicted to be closely linked to the production of specific alkenes, as key components of the sex pheromone mimicry which underpins pollinator attraction. Relative to the closely related *Platanthera* orchid genome, the findings confirm extensive gene duplication, with such novel genes plausible candidates for pollinator driven selection. Without the extensive prior work on the chemistry and biosynthesis of sex pheromone mimicry, these key insights from genome are unlikely to have been found. Finally, the outcomes of an innovative population genetic study of *O. sphegodes* and three closely related sister taxa is reported. Despite, extensive genome wide similarity and high levels of allele sharing, a 20 mb 'genomic island of speciation' has been found between *O. sphegodes* and its sympatric sister *O. exaltata*, which warrants more detailed investigation in the future.

In conclusion, this study highlights the value of building genomic resources to complement even a well-studied wild species. Furthermore, as the largest orchid genome to be sequenced to-date, the study demonstrates the potential to now tackle the genome sequencing of many other orchids of this genome size and larger.

Despite my great enthusiasm for this study, I was surprised by the claim on line 142 of 'the unique pollination system'. To the contrary, sexual deception is a worldwide phenomenon with multiple

evolutionary origins across the Orchidaceae. Indeed, hundreds of orchid cases are known from across four continents (see Current Biology 33 2023 R453-R518 for a recent plain English review), with species radiations potentially even larger than found in Ophrys. Intriguingly, only two cases of sexual deception are known from outside the orchid family, begging the question of why this strategy is far more prevalent in the orchids. Building on this first genome case study, high quality genomic resources across a diversity of sexually deceptive orchids could help unlock the key to this and many other seemingly mysterious aspects of sexual deception.

I can only assume this claim of 'unique pollination' was an unintentional oversight, which can be easily rectified with a small expansion of the text, and one or two additional supporting references. Furthermore, there is a missed opportunity to set up for future studies some genomic hypotheses that might apply to the repeated evolution of sexual deception and the very frequent evolution of specialised pollination in the orchids more generally.

Below I offer some additional comments, suggestions and queries.

Ln 55, 60. In order to provide a balanced perspective, I suggest a short sentence with supporting reference(s) that notes that there is some literature challenging the high degree of specificity and the number of Ophrys species. Perhaps something like: "Notwithstanding some uncertainty about the number of species and the extent of pollinator sharing, at the local population level extreme specialisation is evident in most Ophrys species".

Ln 86. Suggest or expanding sentence here as in the Supplement to include 'function, gene ontology (GO), and protein domain ...'

Ln 94, 'novel sequences' or 'novel sequence motifs'?

Ln 104, suggest it would be helpful to add a short plain English sentence on how this age estimation was done. Alt. give the assumptions used to make the estimates.

Ln 185, suggest it would be helpful to annotate SAD gene numbers around the circle of Fig 3, for reader ease. Otherwise, the reader is forced to infer location from the terminal labels, which are quite hard to read.

Ln 200, is '15 gene models' correct? Suggest, '15 of the 16 FAR homologues form a phylogenetic clade ...'

Ln 221, suggest replacing 'majorly' with 'make a major'.

Ln 267, Suggest it would be helpful to the reader to spell out the final number of individuals used for sequencing.

Finally, based on the mss alone, I was confused as to when Illumina sequencing was used. If I understand correctly from the supplement, Illumina sequencing was used to generate additional short-read libraries, and for the Hi-C. If so, I suggest minor changes in the main text to clarify:

Ln 73/74. Suggest adding 'sequenced with Illumina'

Ln 81/82. Suggest adding "Hi-C" Illumina sequence reads.

Ln 284-292. Please spell out use of Illumina sequencing more clearly.

Rod Peakall
The Australian National University

Reviewer #3:
Remarks to the Author:
Dear Authors:

I enjoyed reviewing your paper "The genome of the early spider-orchid *Ophrys sphegodes* provides insights into sexual deception and adaptation to pollinators". I found it to be clearly written and interesting.

I believe this is a valuable paper for the pollination field, especially for systems relying on sexual deception which has not received as much attention in terms of their genetic architecture as other pollination systems. Having a genome of *Ophrys* available and identifying the likely means by which it has adapted and radiated over a short period of time is a breakthrough.

I found the methods you provided clear and comprehensive. The results are sensible and clearly interpreted, which will make this paper valuable to anyone interested in sexual deception. It was interesting to note the dominance of repetitive elements and the role they have played, and it would be great to see if this matches radiations in other sexually deceptive plants.

The figures are mostly clear, although I would suggest a few amendments. You might consider removing the number of expanded and contracted gene families on the branches in Fig. 2a, as the bubbles at the tips clearly depict this information and will make for a cleaner image. It would also be useful to add a thumbnail photo of a flower of *O. sphegodes* and *P. zijinensis* in Fig. 2b (as you have done in Fig 4b). For Fig. 4, naming each species and their abbreviations/colours at the start in Fig. 4a would improve clarity as this information currently only appears at the end of the legend. It may be even better to include it in the figure itself, as readers will likely view this before reading the full legend.

To enhance the range of this paper, it would be great to include a bit more context about sexual deception and where it occurs. I realise there are space limitations, but some mention here of other sexually deceptive systems outside of Europe that have also undergone recent diversification in response to becoming sexually deceptive would be informative. Adding a more global context of this pollination strategy will position this paper better within the field, especially considering your focus on sexual deception.

A stronger focus on pollinators, in particular the behaviour of deceived pollinators and the selection they exert on sexually deceptive orchids, would help support the conclusions and claims made here, especially considering your use of "adaptation to pollinators" in the title. Adding these angles will also help to highlight the novelty of this paper.

I appreciate the inclusion of a phylogenomic tree comprising several angiosperms for which full genomes are available. This is a welcome addition and should also be of interest to readers beyond those working on sexual deception.

Point-by-Point Response to Reviewers' Comments

First, we would like to thank all three reviewers for taking time to review our manuscript and for providing positive feedback and constructive suggestions that helped us to improve the manuscript. Below, please find a point-by-point response to the reviewers' comments with our responses in blue and with comments numbered unambiguously for clearer communication.

Reviewer #1

Russo et al. reported the chromosome-scale genome sequence of *Ophrys sphegodes*, providing evidence for chromosomal rearrangements in the *Ophrys* lineage. They also identified a transposable element expansion event in the *O. sphegodes* group and highlighted the role of gene duplication in contributing to the evolution of chemical mimicry. In addition, the authors proposed a candidate genomic region associated with pollinator-mediated evolution. I have read the preprint of the manuscript in October and found it intriguing. However, some major revisions are necessary for it to be suitable for publication in Nature Communications.

[R1.1]

1. This work conducted whole-genome sequencing to unravel the mechanisms underlying sexual deception in *Ophrys sphegodes* and its adaptation to pollinators that fosters reproductive isolation and species differentiation. The manuscript centers around genomic sequencing but falls short of fully utilizing genomic datasets to address significant questions:

Response:

Thank you for your assessment. We agree that in our original manuscript we had not used the resources we created to the full extent possible. In response to Reviewer 1's comments, we have carried out numerous changes to the manuscript and, to the extent that this was possible, added several new analyses. We do think this has improved the paper greatly and are grateful for Reviewer 1's input. We shall go through the individual comments below.

[R1.1.1] (1) Despite the challenge posed by the highly-modified nature of sexually deceptive flowers, it's suggested to perform genome evolution and comparative analyses to investigate the origin of sexual deception pollination in *Ophrys*.

Response:

*This is a good point (also echoing comments by reviewers 2 and 3) and one that we agree with. The limitation is that we only have the genome of *Ophrys*, which sits on a long branch in the Orchidoideae phylogeny, whereas no genomes are available for the related and non-sexually deceptive genera (such as *Steniella*, *Serapias*, *Anacamptis*, *Himantoglossum*). Without a genome from any of those genera, we are not in a position to perform genome analyses that might reveal the specific changes that occurred in the shift towards sexual deception that must have occurred alongside the evolution of the genus *Ophrys*. We are convinced that in the next years more genomes will be released, so that this analysis will hopefully soon be possible.*

With this caveat aside, there are some relevant comparative analyses that we performed. Given that odour-based attraction of pollinators is crucial in sexual deception and that this differs from the other orchid genera, we have leveraged the fact that we had previously identified candidate genes for this trait (Tables S7 and S8) and gene family analyses in order to combine these two for a comparative analysis of (a) hydrocarbon (VLCFA-based odour) biosynthesis, (b) anthocyanin biosynthesis and (c)

floral perianth MADS-box genes presented in the new Figures, Fig. 3, Fig. S13 and Fig. S11 respectively. While this again illustrates the expanded gene families of SAD, FAR, FAT, CHS in *Ophrys*, the lack of more closely related orchid genomes does not allow us to decide if these expansions are *Ophrys*-specific or if they were already ancestrally present in the lineage out of which *Ophrys* evolved.

[R1.1.2] (2) Metabolic data should be combined with whole-genome data to reveal the sex phytohormone pathway in *Orchrys*.

Response:

This ties in with the previous comment R1.1.1. We assume that Reviewer 1 meant to show pathway maps, which we have now added for hydrocarbon and anthocyanin biosynthesis (Fig. 3, Fig. S13). In line with comments R1.2.3 and R1.1.3, we also show RNA-seq expression data for these pathways (see response to R1.2.3). The data are limited in that they only provide information on fully developed labella but not on earlier stages that may be relevant for a more detailed functional understanding.

For the hydrocarbon biosynthesis pathway that gives rise to the pseudo-sex-pheromone (alkenes), the findings are broadly consistent with previous findings, although the role of some genes showing differential expression that we had previously be unaware of, will merit future analysis (one FAD, one KAS, one KCS, two FARs).

For the anthocyanin biosynthesis pathway, the available data from open flower labella highlights two CHS homologues, but since this is at the base of the pathway and hence potentially relevant for functions outside of pigment production, we cannot offer a satisfying explanation at this point. As regards anthocyanin pathway genes, Sedeek et al. (2013, PloS One) had not been able to identify a F3'5'H homologue in the *Ophrys* transcriptome and hypothesised that this gene may have been lost. Indeed, we have not been able to identify a homologue in the genome. We note this point in the figure legend.

[R1.1.3] (3) Leveraging the whole-genome data in conjunction with population genetic structure to elucidate the functional differentiation mechanisms of sexual deception in *Ophrys* species.

Response:

We share the curiosity to answer this complex question and several of the labs contributing to this manuscript have long-standing research programmes to investigate these aspects. Needless to say, we can only address selected aspects of this question in the present manuscript. Given the finding of a strongly differentiated region on chromosome 2 (related to comment R1.2.4) that primarily separates *O. sphegodes* and *O. exaltata*, we here place our emphasis on the differentiation between these two species.

Firstly, intra-population and intra-individual genetic diversity is high in this outcrossing plant, whereas genome-wide 'population genetic structure' in the classical sense is almost absent or only very weakly detectable in the *O. sphegodes* group (see Fig. 5, S16 and e.g., Soliva & Widmer 2003, *Evolution*; Mant et al. 2005, *Evolution*; Sedeek et al. 2014, *Mol. Ecol*; Cozzolino et al. 2020, *J. Syst. Evol.*), with most alleles shared across species - as is expected for early stages of genic ecological speciation - and only few highly differentiated loci (discussed in more detail e.g., in Sedeek et al. 2014, *Mol. Ecol.*). This was why we carried out an F_{ST} -based analysis, which essentially confirms Sedeek et al.'s (2014) findings, and confirms their prediction of more-than-random linkage among significant F_{ST} outlier loci – essentially we can see that most of them are on chromosome 2. For the sake of brevity and readability, we do not discuss all of these details in the manuscript because we essentially confirm previous analyses. Instead, we focus on the novel aspect of spatial clustering of these highly differentiated markers (which are mostly on chromosome 2) that conspicuously looks like an 'island of speciation'. (Discussion of the few other elevated loci in the genome has been omitted for the sake of brevity and because they do not really form similar 'islands'.)

Secondly, looking for functional mechanisms that explain species differences is not trivial for the aforementioned reasons and requires extensive data sets. So far, this has been carried out for SAD genes (Xu et al. 2012, *Plos Genet.* in conjunction with Xu & Schlüter 2015, *Ecol. Evol.* and Sedeek et al. 2016, *Curr.Biol.*), which show a complex pattern of copy number, allele, expression and functional variation (different alleles may differ in their enzymatic activity). While the data from the *Ophrys* genome are consistent with these previous data (see e.g., confirmation of expression patterns in Fig. S15, they do not provide more functional insights (but confirm gene duplication events). These SAD genes may not fit simple models of biallelic, single-locus divergent selection, which makes it harder for them to be picked up in an F_{ST} scan. Additionally, our population genomic scans are currently limited to detecting differentiation in coding sequences, yet it is clear that some of the differences in SAD genes are at the level of expression. As (despite efforts to identify them) we currently do not know any specific cis- or trans-regulatory sequences or genes, our population genomic approach is currently limited in this respect.

In the revised manuscript, we include several additional analyses, e.g., confirmation of the GBS-based F_{ST} analysis with independent RNA-seq data (please see response to point R1.2.4 for details) and we add a table of genes with/near SNPs with elevated F_{ST} (Table S13 for chr. 2).

[R1.2] 2. Most of the discussion seems speculative without providing robust analysis and evidence:

Response:

We wish to respond to this general point here before individually addressing the specific examples of Reviewer 1 below. Reviewer 1 is certainly correct that it is challenging to provide conclusive evidence of past evolutionary processes and that part of the discussion must therefore necessarily be cautious and might even be speculative. However, we do think that a discussion that raises hypotheses which can be addressed by the community is valuable and appropriate. The hypotheses we raise are not taken out of thin air but are (we think) plausible and relevant to the current discussion of TE-driven evolution in the field¹; hopefully they will contribute to stimulating further research.

Having said this, we recognise that we should have been more careful in our wording, so that the reader is better able to recognise what is straight inference and where we raise a hypothetical scenario. We have therefore toned down the discussion in several places and rephrased some passages for improved clarity in this respect (see specific points further down).

As regards analyses, while we have carried out a set of additional analyses on the *Ophrys* genome, many of them yield complex results; it would be premature to depict them as supporting specific evolutionary scenarios with high confidence. It is clear that it will take several years of effort by the community to fully understand the detailed processes of evolution in this complex non-model organism. Below we address specific points raised by Reviewer 1.

[R1.2.1] (1) In the section titled “A burst in transposon activity preceded adaptive radiation”, the authors suggested that TE insertions led to the adaptation of new pollinators, relying solely on literature evidence without presenting results supported by original analysis. It is also unclear how the TE burst enabled *Ophrys* to adapt to a new pollinator niche and facilitate adaptive radiation, which is an important new insight of the manuscript.

¹ We shall not include a full review of the literature here, but the following papers give some impression of the ideas in the literature, many of which are related to the hypotheses we raised: Carducci et al. (2019) *Eur. Zool. J.*, doi: 10.1080/24750263.2019.1695967; Lee et al. (2017) *Sci. Rep.*, doi: 10.1038/s41598-017-08194-5; Martelossi et al. (2023) *BMC Biol.*, doi: 10.1186/s12915-023-01632-z; Oggenfuss (2021) *eLife*, doi: 10.7554/eLife.69249; Shao et al. (2019) *Sci. Rep.*, doi: 10.1038/s41598-019-51888-1; Sotero-Caio (2017) *Genome Biol. Evol.*, doi: 10.1093/gbe/evw264; Wong et al. (2019) *PNAS*, doi: 10.1073/pnas.1910106116

Response:

We did not actually claim that TE insertions directly led to the adaptation to new pollinators, but provided a possible scenario how TEs may have contributed to allowing pollinator adaptation to happen. As regards Reviewer 1's question on **how** TE expansion may have enabled adaptation to (or diversification via) new pollinators, considering that *Andrena/Colletes* pollinator attraction is strongly dependent on olfactory cues (which have been proposed to be the first to diverge during speciation; Sedeek et al. 2014), this is a question of how TE insertions may affect hydrocarbon biosynthesis and lead to the production of new odour bouquets (e.g., via new compounds or new mixtures of existing ones). While our knowledge of (especially) regulation of hydrocarbon biosynthesis in *Ophrys* is limited, an example of how novel metabolites can evolve is given in nicotine biosynthesis of *Nicotiana attenuata*; here, TE insertions distributed TF binding sites into the regulatory regions of genes that originally belonged to different metabolic pathways, which led to their co-expression and the formation of the novel nicotine biosynthetic machinery.

To summarise this briefly, we have edited the following passage from

[original] “The ability to cope with genome size changes has allowed angiosperms to successfully diversify³² and TEs have played an important role in enhancing angiosperm evolution³³ through their effect upon gene expression³⁰, as well as gene duplication and genomic rearrangements^{34,35}. By contributing to the generation of intraspecific genetic diversity³⁶, TE bursts may have provided the *O. sphegodes* lineage with the genetic capacity to adapt to a new pollinator niche (i.e., *Andrena* bees), thereby facilitating the adaptive radiation of the *O. sphegodes* group.”

To [revised, line 128ff.] “ The ability to cope with genome size changes has allowed angiosperms to successfully diversify³² and TEs have played an important role in enhancing angiosperm evolution³³ through their effects upon gene expression³⁰, as well as gene duplication and genomic rearrangements^{34,35}. Since TEs often carry transcription factor binding sites, TE expansion can rewire existing metabolic networks and facilitate the evolution of new compounds (or mixtures), as it was shown for the evolution of nicotine biosynthesis in tobacco⁴⁴. Thus, TE bursts can contribute to the generation of intraspecific genetic and metabolic diversity³⁶. Since changes in pollinator-attractive hydrocarbon compounds are suspected to be involved early in speciation in the *O. sphegodes* lineage¹¹, it is tempting to speculate that TE bursts may have provided this lineage with the genetic capacity to adapt to a new pollinator niche (i.e., *Andrena* bees), thereby facilitating the adaptive radiation of the *O. sphegodes* group.”

We have manually checked TE insertion age and genome structural features around a small number of biosynthetic genes of specific interest so far (including the *SAD1/2/7/8* locus), but this analysis is limited and has not revealed conclusive answers; therefore, we clearly declare the speculative nature of the concluding statement above. Please also see our response to related comment R1.4.4.

[R1.2.2] (2) For the discussion on “Expanded gene families may be involved in flower development”, the evidence is based solely on a paper published in 2013. In addition, only in the supplementary 10 showing the phylogeny of MADS-box and MYB genes. However, I do not understand each of the subfamily represent what kind of MADS-box gene as well as MYB genes. The authors also indicated LOB domain TFs, C2C2-GATA, WRKY, TCP, YABBY, NAC TFs might involve in development. However, actually many of these TFs not only involved in the development but many other aspects of biological processes, such as biotic and abiotic stress responses. It is recommended to conduct analyses of functional genes based on genomic evidence and add analysis on flower morphogenesis and pigmentation, which is key to provide evidence for how flower morphology and color simulate the perception of pollinating insects.

Response:

We share Reviewer 1's desire to delve deeper into the mechanisms and have conducted several follow-up analyses and made changes to the text. There are several parts to this comment that we respond to separately:

*(i) We reject the notion that our evidence in this section is based 'solely' on the 2013 paper by Sedeek et al. (previously Ref.50, now Ref.64) In fact, it is only cited once here, in the penultimate sentence of the entire section. This paper, however, had used transcriptomic and proteomic data to identify candidate genes of interest for pollinator attraction in *Ophrys*, focusing primarily on hydrocarbon and anthocyanin biosynthesis. We therefore made sure to trace down those previously identified candidates in our genome data (all of these were manually validated, Table S7) and we spent considerable time and effort identifying additional genes of interest in our study. To be clear, FAR and CHS had previously been put on the radar of genes of interest by Sedeek et al. (2013), but the inference that these gene families are expanded comes from our own analysis. In order to clarify this, we have inserted the words 'previously identified' into the sentence:*

[revised line 194ff.] " Finally, two previously identified putative candidate gene families for floral odour production and anthocyanin biosynthesis⁵⁰ showed significant expansion, too, namely fatty acyl-CoA reductases (FARs; Fig. 3) and chalcone synthases (CHSs; Supplementary Fig. 13), also involved in defence response (29 and 14 genes, respectively)."

(ii) Myb and MADS-box genes in [old] Fig. S10 – thank you for noticing this. Indeed, these trees were lacking annotation information. We have updated the figures correspondingly. For better readability, we separated the two trees into two figures; due to other changes in the manuscript, they are now Supplementary Figures 10 (MADS) and 12 (Myb). Description of the MADS-box gene annotation has also been updated in SI section 2.4.

*We note that, as indicated in the main text, the majority of MADS expansion occurred in type I MADS genes, whose function is less well understood. As regards 'classical' ABC MADS genes, the copy numbers are very similar to those in other orchids (Fig S11), including *Orchis italica*, for which no genome is available, but which is the species most closely related to *Ophrys* for which studies on floral development have been carried out (Valoroso et al. 2019).*

*(iii) Reviewer 1 is correct that the listed TFs may also be involved in functions other than development, including stress responses. Sadly, detailed studies on these gene families are largely lacking for *Ophrys* and related plants (but see e.g., Lucibelli et al. 2021 Int. J. Mol. Sci. on orchid YABBY genes), so that we refrain from discussing these TF families in more detail here. Given the working hypothesis that at least some of these may be broadly conserved and function similarly as in *Arabidopsis*, however, members of all of the stated families carry out functions that would be essential in producing the complex labellum of a plant like *Ophrys*. These gene families will therefore very much be on our radar in future studies, even if we do not discuss the issue further in the manuscript. In the revision, we have toned down the wording in the section title and added the mention of stress response/other processes in the sentence at line 185 as follows:*

"... were genes encoding transcription factors (TFs) involved in plant reproduction and flower development (but also other processes, e.g., stress responses), such as ..."

Likewise, we added 'for instance' into the sentence at line 185; it now reads:

"WRKY TFs (11 genes) involved for instance in trichome development ..."

*(iv) 'evidence for how flower morphology and colour simulate the perception of pollinating insects' – such studies have previously been carried out for *Ophrys* (older literature reviewed by Schlüter & Schiestl 2008. Trends Plant Sci.). This includes studies on the function of trichomes (Ågren et al. 1984, Nova Acta Reg. Soc. Sci. Ups.), perigon coloration (which increases colour contrast and visitation rates in some species; Streinzer et al. 2010 Athr.*

Plant Int.; although probably not for *O. sphegodes* s.l.; Vereecken & Schiestl 2009 *Ann. Bot.*) or UV-reflective speculum patterns that increase pollinator attraction and can then be memorised by pollinators (avoidance learning; Stejskal et al. 2015, *PLoS One*). For the studied species specifically, Sedeek et al. (2014, *Mol. Ecol.*) showed that there is a rough correspondence between bee body coloration and labellum, a variable UV-reflective speculum pattern (against a dark, achromatic labellum background) and differences in 3D geometry between species that would affect pollinaria placement on a pollinator's body. Key points of these previous studies are summarised in the revised manuscript (please see response to comment R3.4).

(v) 'add analysis on flower morphogenesis and pigmentation' – While we agree that this would be nice, it is not possible for us to do here. Studying floral development (including morphogenesis and pigmentation) is, in fact, a massive undertaking in a plant that cannot be grown easily in the lab, has no genetic toolkit available, in which flowers largely need to be collected in the wild, and for which no basic data on flower development are available. We have initiated a PhD project (in the lab of PMS) to study exactly this, but the project is still ongoing and we do not currently have such data.

(vi) follow-up analyses: We have conducted several new analyses that we present in the revised manuscript: comparative analysis of gene copy number among orchid genomes for (a) hydrocarbon biosynthesis, (b) anthocyanin biosynthesis and (c) floral MADS-box genes. In addition, we performed differential gene expression analysis from RNA-seq data (Piñeiro Fernández et al. 2019, previously used only for transcriptome assembly and phylogenetic analysis) mapped against the *Ophrys* genome. These data are derived from mature labella of unpollinated flowers. Given that the most interesting finding of our genome-wide population genomic analysis was differentiation in chromosome 2 that appeared to be primarily between *O. sphegodes* and *O. exaltata*, we focused our expression analysis on these two species. These analyses are presented in the [new] Fig. 3, Suppl. Fig S11 and S13. They do however, not identify any novel processes or gene functions beyond what we had previously presented. If anything, the finding that floral MADS-box gene (Suppl. Fig S11) copy number appears to be similar to other orchids (and expression in line with *Orchis*), suggests that the stark differences between flowers likely develop downstream of these genes. We inserted a comment to this effect in line 188ff., which reads “The configuration of MADS-box genes putatively involved in perianth specification in *Ophrys* appears similar to other orchids (Supplementary Figure 11), including the related *Orchis italica*⁶³, suggesting that the stark difference in their flowers likely results from the action of downstream genes.”

[R1.2.3] (3) Again, in the section titled “Plant adaptation to pollinators in *Ophrys* is associated with local duplication of candidate genes”, additional transcriptome evidence is needed to support the duplication genes.

Response:

We are not entirely sure what Reviewer 1 means here, but we are confident that gene duplication levels are not high due to artefacts and that, to the extent that we have been able to check this manually, the duplications in the candidate genes are real. We have used transcriptome data (previous transcriptome assembly + PASA pipeline; plus independent RNA-seq reads) as well as PacBio data to check the assembly. Please see our response to point R1.3, where we discuss this in more detail.

If Reviewer 1 is asking for gene expression data, then we must note that unless all tissues and stages of a plant are included, failure to map reads to a putatively duplicated gene would not imply that the gene does not exist or is not active at all. With this caveat in mind, we have used RNA-seq data available for the *Ophrys sphegodes* group (raw data from Piñeiro Fernández et al. 2019 but with an entirely novel analysis) to analyse gene expression and differential expression in all pathway candidate genes between *O. sphegodes* and *O. exaltata*. The sequence data are only from mature (but not from developing) flower labella, so do not cover the entire range of tissues and stages of interest.

However, the majority of hydrocarbon and anthocyanin candidate genes are expressed in these labella; those that are not (3 CHS, 1 F3'H, 1 AGXT, 1 DFR for anthocyanins; 4 ACP, 2 FAD, 1 FAT, 1 LACS, 3 FAR for hydrocarbons) do not have any indications in their sequence that would suggest they are not plausible genes. The possible exception is ACP (acyl carrier protein), a very small, single-exon protein, and sanity-checking of such short sequences is comparatively difficult. Certainly, for the more diversified gene families (e.g. CHS, FAR, etc.), one may well expect there to be subfunctionalised copies with different expression patterns, which might provide a reason for not observing expression for these genes in the tissues available in the data set.

Overall, the expression data suggest that, by and large, the candidate duplicated genes are real. Since the level of differential expression between species in these pathway genes is low in mature labellum tissue, more detailed data sets will have to be collected to gauge their potential role in species differences. In the case of SAD genes, there is substantial previous work, so that we have been able to compare previous expression data with the new RNA-seq based expression pattern in more detail (Fig. S15), taking into account the complicated sequence relationships (compare notes in Table S7 and Xu et al. 2012 Plos Genet.). In short, our expression data are in line with the larger, RT-PCR-based data set of Xu et al. (2012; their Fig. S3), showing that SAD5-type sequences potentially involved in alkene synthesis (SAD5,6,9,10) are expressed strongly in *O. exaltata* with low or absent expression of SAD2-type sequences (SAD1,2,7,8), whereas the opposite is true for *O. sphegodes* (high SAD2-type, low SAD5-type). We very briefly indicate this in line 218 by stating:

“It is also consistent with expression patterns revealed by RNA-seq data (Supplementary Fig. 15).”

Also, to ensure we do not oversell the data presented, we toned down the wording in the section title (“may involve” instead of “is associated with”).

[R1.2.4] (4) In “Population genomics analyses reveal putative barrier loci under pollinator-driven selection”, there seems to be inconsistency between the discussion of chr2 and the results shown in Fig3 (Chr4). Please further analyze the highly-differentiated c. 20Mb region on Chr2, including genome annotation and functional enrichment analysis. It is important to show whether key genes related to sexual deception are located in this region and, if not, provide an explanation for how this region contributes to pollination isolation.

Response:

We agree with Reviewer 1 that this is an important point that had not been discussed sufficiently in the previous manuscript. Indeed, the obvious candidate genes we analyse (such as the SAD2-locus on Chr4 in [now] Fig. 4) are not located in the differentiated region of interest on chromosome 2.

We incidentally realised that the statement regarding the location of this island of differentiation in the main text was incorrect (it is 333-352 Mb on Chr2, not as originally stated 327-346 Mb). We have corrected this error.

To address this reviewer comment (and related to point R1.1.3), we carried out additional analyses with the aim to figure out which molecular processes might contribute to species differentiation. The bulk of these data are shown in the new Fig. S18.

Firstly, we aimed to confirm the differentiated region with independent data. To this end, RNA-seq data (raw data from Piñeiro Fernández et al. 2019) were mapped and variant-called as detailed in the Supplementary Materials. The data set contains few (6-8) individuals per species for *O. sphegodes*, *O. exaltata*, *O. garganica* and *O. incubacea*, i.e., the same species as the GBS data set. While the GBS dataset has more individuals per species and hence estimates species allele frequencies better, the RNA-seq data set has higher coverage and samples the genome much more densely. We then used these variant calls for carrying out F_{ST} analysis similar to what we did with the GBS data set. Given that the chromosomal region of interest appeared to differentiate *O. sphegodes* and *O. exaltata*, we present results from both GBS and RNA-seq data sets for both a ‘global’ analysis (all 4 species) and a

pairwise analysis between *O. sphegodes* and *O. exaltata*. (Chr2 data in Table S13 and plotted in Fig. S18).

These analyses confirmed (i) that the majority of elevated loci from the ‘global’ GBS-based (and also RNA-seq) analysis are also strongly differentiated between the two focal species; and (ii) the region of differentiation on chromosome 2 (333-352 Mb). However, with the denser sampling across the chromosome that RNA-seq data afforded, a larger interval containing the original region can be defined. Using the first and last elevated F_{ST} marker ($F_{ST} > 0.25$; this threshold is arbitrary, but changing it does not affect the interval identified), the region as defined RNA-seq data is ~327-358 Mb (roughly 30.6 Mb, or 0.6% of the genome). Further analyses have therefore been conducted using both definitions of this ‘island’.

We summarise this as follows:

[lines 272ff] “In an independent F_{ST} analysis based on RNA-seq data, this genomic region was confirmed as being differentiated between these species, although this dataset offers denser sampling of the genome and yielded a somewhat larger interval (327 – 358 Mb, 31 Mb in length; 0.6% of the genome; Supplementary Fig. 18; Supplementary Table 13). ”

It would be highly interesting to know if this is a region genuinely associated with species differentiation (as in a ‘speciation island’ caused by divergence hitchhiking) or simply reflects a large inversion that suppresses recombination. We used Illumina WGS data from other species, including *O. exaltata* (coverage ~10x), to carry out structural variant (SV) calling with three different software pipelines (Delly, Manta, Lumpy), considering variants called by at least 2 out of 3 SV callers. Shown in the graph below are inversions on part of chromosome 2 (300-360 Mb):

The graph shows that there are only 4 inversions in *O. exaltata* > 50 kb. The largest inversion called overall (including in *O. exaltata*) is just over 500 kb. Overall, we find no evidence of a large inversion that could explain the region of high F_{ST} differentiation. However, with only Illumina data of relatively low coverage and no long-read data to back this up, we consider this analysis to be underpowered to detect inversions and hence inconclusive. We have therefore decided not to include these analyses in the revised manuscript. Our statement in the discussion [line 275ff.] “Whether this pattern of differentiation is due to divergent selection and suppression of effective recombination via, e.g., divergence hitchhiking or an inversion, is currently unknown” stands effectively unchanged.

Secondly, we performed GO enrichment analysis for the differentiated region (using both the GBS- and RNA-seq-based definition). The result of the top GO terms (with $p < 0.01$) are shown in Table S14. Here, the top biological process (BP) term is GO:1903415 (flavonoid transport from endoplasmic

reticulum to plant-type vacuole') with 4 genes present in the region of interest. Three of those are homologues of *Arabidopsis* TRANSPARENT TESTA 9 (TT9), a peripheral, Golgi-localised membrane protein involved in membrane trafficking, vacuole development and the accumulation of flavonoids/anthocyanins in the vacuole (Ichino et al. 2014). We summarise this as follows:

[lines 277ff.] "Since the differentiated region of chr 2 did not contain any *a priori* candidate genes for hydrocarbon or pigment biosynthesis (Supplementary Tables 7 and 8), we performed a GO enrichment analysis (Supplementary Table 14). This revealed term GO:1903415 ('flavonoid transport from endoplasmic reticulum to plant-type vacuole') as the most significantly enriched 'biological process' category. Three genes (*Osph2G68830*, *Osph2G68850* and *Osph2G68960*) with this term are homologues of *Arabidopsis* TT9 (AT3G28430), a peripheral membrane protein necessary for vacuole development and the accumulation of anthocyanins in the vacuole⁷⁹."

Thirdly, we screened all genes in the region of interest (spanning 661 genes when using the GBS and 1153 when using the RNA-seq definition) for indications that they may differ between *O. sphegodes* and *O. exaltata*. We screened for:

- (i) genes containing SNPs with elevated F_{ST} (>0.25 ; new table S13; Fig. S18a), using the pairwise F_{ST} between the two species;
- (ii) genes annotated as transcription factors (Fig. S18a);
- (iii) genes with excess amino acid nucleotide diversity (i.e., the non-synonymous exceeding synonymous nucleotide diversity, that is $\pi_n - \pi_s > 0$; we chose to use the difference rather than the ratio as this is more straightforward to visualise; Fig. S18c)
- (iv) genes with differential expression in a genome-wide analysis between the two species ($FDR < 0.05$)

We then looked for overlaps in the sets of genes identified in these four analyses. This yielded only very few candidates (Fig. S18d):

- the most interesting were two transcription factors that also had elevated F_{ST} , namely an B3-ARF (auxin response factor) and an AP2 homologue;
- another gene, a putative ubiquitin conjugating enzyme, showed both amino acid change and differential expression
- the remaining 4 genes had no informative annotation information available.

Overall, while we cannot be certain that any of the genes identified here is directly involved in differentiating *O. exaltata* from *O. sphegodes*, it would seem more likely that regulatory, rather than biosynthetic processes, are at play.

We summarise our findings as follows:

[lines 285ff]: "Additionally, we screened genes in this region for (i) elevated F_{ST} between *O. exaltata* and *O. sphegodes*, (ii) annotated transcription factors, (iii) genes with excess amino acid change (i.e., excess non-synonymous nucleotide diversity), and (iv) differential floral labellum gene expression between the two species (Supplementary Fig. 18). Only 7 genes were found in more than one screen (Supplementary Fig. 18d), three of which had useful annotation information that revealed one AP2 and one B3-ARF TF (*Osph2G66470* and *Osph2G63210*, respectively) as well as one ubiquitin conjugating enzyme (*Osph2G63100*). While it remains to be established whether any of these genes are directly linked to differences between the two *Ophrys* species, currently it seems more likely that regulatory rather than biosynthetic genes contribute to this genomic region of differentiation."

[R1.3] 3. Quality and assessment of sequencing data. It is necessary to further confirm the following genome assembly and annotation results: The assembled contig N50 value (754kb) appears to be relatively low compared to current sequencing technologies for genomes. Can this be improved? The BUSCO values are acceptable, while this may be attributed to the inherent characteristics of Nanopore sequencing. Nanopore sequencing may also result in elevated annotated protein-coding genes compared to other sequenced orchids. This overrepresentation includes an excess of MADS-box genes and transcription factors (TFs), as well as the duplicated genes, which need to be further confirmed.

Response:

While we agree with Reviewer 1 that scepticism is in order, we do think that the overall quality of our genome assembly and annotation is high and that the key results are not artefactual. There are several points linked to this reviewer comment, which we shall discuss individually below:

(i) N50 value - Notwithstanding the fact that N50 values achievable using the newest long-read technologies are more impressive than our initial value of 754 kb, we think that our N50 is good when considered in context. Firstly, our initial N50 values (assembly Osph-v0.7) can be attributed to the type and quality of data that was used. The raw Nanopore data were generated in 2018, when Nanopore data quality was not as high as it is now (although we repeated base-calling at a later stage with improved software and polished Nanopore reads with high-accuracy Illumina data as detailed in Supplementary Methods paragraph 1.2). Given this was a very challenging project overall, we were not in a position to submit this manuscript at a time when our initial N50 value would have sounded more impressive – e.g., Table 1 in Chung et al. (2021; Mol. Ecol. Res., doi: 10.1111/1755-0998.1353) shows the N50 values of then-available orchid genomes with the sole exception of *Cymbidium goeringii* to be less than 100 kb. In this context, we consider our initial N50 of 754 kb based on long- and ultra-long (up to 1.7 Mb) Nanopore reads to be more than adequate for the first rough draft stage of assembly (Osph-v0.7). Genome continuity was then improved in several steps, including Hi-C data, to yield a chromosome-level assembly with N50 of 218 Mb (L50 of 10; Table S4). To our judgement, these statistics are impressive considering the challenges of assembling such a large and heterozygous genome from limited amounts of available input material.

(ii) We are aware that a low-quality genome assembly can result in overrepresented annotated genes. To avoid this issue, and also to deal with heterozygosity, we removed under-collapsed heterozygous contigs during the genome assembly steps. This ensured the removal of duplicated (or heterozygous) contigs that could inflate genome size and gene number (Redundans step in SI section 1.3). To make sure that the annotated genes did not include artefacts, we performed several filtering steps as described in SI section 2.2, including the removal of gene structures predicted multiple times, the retention of genes with RNA-seq evidence or functional annotation. Finally, we leveraged previously independently assembled transcriptomic data to correct gene number and structure (PASA step in SI section 2.2). If gene expansion were an artefact due to poor genome quality, we should see overrepresented genes homogeneously in most of the gene structures. Instead, many genes are single-copy genes and many gene duplications appear to have been in tandem and localised to specific regions. This is evident for example in the candidate of genes involved in the anthocyanin and hydrocarbon biosynthetic pathways (Fig. 3 and S13), manually checked, where *Ophrys* does not consistently have the highest gene copy number as compared to other orchids. Similarly, floral MADS-box genes (new Fig. S11) are in the expected range of gene copies, although there has been expansion in some specific clades of MADS-box genes (e.g. many type I Ma MADS copies on Chr 12 and Chr 13, now annotated in the revised Fig. S10).

(iii) We also note that based upon numerous manually checked genes (anthocyanin, carotenoid, hydrocarbon biosynthesis genes as well as MADS and Myb-type transcription factors), we think that the majority of under-collapsed contigs removed with the Redundans pipeline (SI section 1.3) represent genuine alternative alleles at these loci, with plausible alternative sequences and matching intron/exon structure. We are best positioned to judge this for the SAD desaturase loci, because for these we can draw upon a wealth of previous, independent data; there we can find previously sequenced alleles in alternative haplotigs of our assembly, the SAD2 locus on chromosome 4 (containing SAD1,2,7,8) having alternative representations for 3 out of 4 gene models (SAD1, 7, 8) in alternative haplotigs not included in the haploid reference sequence. Since adequate discussion of this topic is (a) complex and (b) not of critical importance for our manuscript, we have – for the sake of readability – decided not include these details in the manuscript (but the FASTA file with alternative haplotigs has been uploaded to the figshare archive). However, to come back to Reviewer 1's point, if many 'duplicated' contigs removed by

Redundans are genuine alternative variants, it follows that the assembly itself is unlikely to have suffered from a high degree of technology-based ‘duplication’ artefacts.

*(iv) We had validated our genome by mapping Illumina data (Table S6), which shows high mapping percentages as expected for a high-quality reference. However, since these data are from short reads, they arguably cannot uncover structural problems, such as potential technical artefacts that appear like gene duplications. In response to Reviewer 1’s comment, we have therefore included an additional validation step in the revised manuscript. Specifically, we mapped an independent PacBio dataset (~2× coverage) of *O. sphegodes* to the genome. This data set had been generated from the same individual as the original Nanopore dataset, but due to the lack of ultra-long reads and the low coverage, had not been used for genome assembly. The percentage of total mapped reads for this long-read data set is 98.06%, with 95.07% of reads being ‘primary’ mappings. These statistics support our view that the genome assembly is of high quality and duplication levels not artefactual. These new results have been included in Table S6 and are now mentioned in lines 94/95. Likewise, SI section 1.5 has been updated accordingly.*

*(v) Finally, our BUSCO results on the genome show a duplicate percentage that is comparable to that of the orchid *Vanilla planifolia*, with a genome size of 736.8 Mb (haplotype A, as stated in Hasing et al., 2020). Likewise, our BUSCO results on the gene set show a percentage of duplicated genes smaller than that of the orchid *Dendrobium catenatum*, with a genome size of 1.1 Gb (Zhang et al., 2017). Hence, we think that the duplication level we report can reasonably be attributed to biological factors rather than artefacts resulting from genome size and technology choice.*

[R1.4]

Minor questions:

[R1.4.1] 1. Lines 26-30 are verbose; a concise statement of scientific significance should conclude line 37.

Response:

Thank you for drawing attention to this. We have shortened the second sentence of the abstract to make it more concise, changing

*“The bee orchids (genus *Ophrys*) mimic their pollinator’s female pheromone, shape and colour to lure male pollinators ...”*

into

*“*Ophrys* orchids mimic female insects to lure male pollinators ...”*

We did not change the first and third sentence, because we think this information is important to the reader and shortening these sentences would, in our view, have impaired their readability.

Additionally, we added a concluding sentence:

*“The *Ophrys* genome, as the first for any sexually deceptive orchid, will prove useful for investigations into the repeated evolution of sexual deception, pollinator adaptation and the genomic architectures that facilitate evolutionary radiations.”*

Other passages have been abridged so as to stay in the word limit for the abstract (now 146 words).

[R1.4.2] 2. The discussion of pollination-mediated evolution needs a more direct link to the title. The cited literatures in lines 53-55 are inconsistent with the genomic heterozygosity (1.28%).

Response:

The first point (link to title) echoes Reviewer 3’s comment R3.4 and our response to this point is detailed there.

Regarding the second point made (heterozygosity), perhaps this is a misunderstanding on part of Reviewer 1, but we disagree that our genomic heterozygosity is inconsistent with the cited literature (Xu et al. 2011, Evolution, and Sedeeq et al. 2014, Mol. Ecol.) and indeed further papers (e.g., Soliva & Widmer 2003, Evolution; Mant et al. 2005, Evolution; Schlüter et al. 2011, Ann. Bot., for *O. fusca* s.l.). In understanding this issue, it is important to realise that the study species, while reproductively isolated by pollinators, are assumed to be in a very early stage of genomic divergence and still harbour a large degree of shared genetic polymorphisms (also see our Fig. 5 for illustration). While speciation is expected to build up genetic divergence between species, this does not imply that we expect the individual within-species heterozygosity to be low. The breeding system is highly **outcrossing**, the orchids displaying high phenotypic variation in traits that pollinators likely use to learn to avoid specific plants (see e.g., Stejskal et al. 2015, PLoS One, for *O. heldreichii*). Due to these mechanisms promoting outcrossing, high amounts of heterozygosity are expected (and observed) in many *Ophrys* species. However, we acknowledge that this may not have been obvious to the reader as we failed to highlight the point on outcrossing in our original manuscript. We have now, along with the changes to comment R3.4, made specific mention of outcrossing in line 61 and hope that this clarifies the issue (please see response to R3.4 for the full text addition covering flower traits and pollinator adaptation).

[R1.4.3] 3. Lines 97-98, the LTR of *Vanilla planifolia* is 10%?

Response:

We double-checked and confirm that this is correct. As stated by Hasing et al. (2020) in their Supplementary Table 2, the percentage of LTRs in Haplotype A of the *Vanilla* genome is 10.0 % (10.1% in Haplotype B). Note that this value is not to be confused with the total amount of repeats, which is of course higher.

[R1.4.4] 4. Lines 104-108, hybridization facilitated by climate events seems speculative without supporting evidence, and lacks a logical connection with reproductive isolation.

Response:

This point is related to related to R1.2.1. and refers to this passage:

“In *O. sphegodes*, analysis of recent LTR insertions shows that LTR activity had an initial increase at around 3 Ma, to reach its maximum at around 1.3 to 0.8 Ma (Fig. 1d). During this period, the Mediterranean Basin experienced climatic oscillations with glacial/interglacial periods^{27,28} that may have acted as environmental disturbances^{29,30} and/or facilitated hybridisation³¹ after distribution range shifts.”

which is then directly followed by the sentence “Any such event might have triggered bursts of TE proliferation in *O. sphegodes*, thus inflating its genome size.”

We obviously cannot know what triggered TE expansion and the intention here was not to make a claim about any specific scenario, but to illustrate that several different candidate mechanisms for TE activation are conceivable that could be direct effects (e.g. temperature) or indirect effects (e.g. hybridisation) of environmental changes that happened at the time. The list could obviously be extended.

We therefore agree with Reviewer 1 that listing specific scenarios like hybridisation (which we incidentally think is plausible and relevant to diversification) is not called for here. We have toned down the passage and removed mention of hybridisation/range shifts. The revised passage reads as follows:

[lines 121ff]: “In *O. sphegodes*, analysis of recent LTR insertions showed that LTR activity had an initial increase at around 3 Ma, to reach its maximum at around 1.3 to 0.8 Ma (Fig. 1d). During this period, the Mediterranean Basin experienced climatic oscillations with glacial/interglacial periods^{27,28}. It is conceivable that such environmental disturbances^{29,30,31} might have led to bursts of TE proliferation in *O. sphegodes*, thus inflating its genome size.”

[R1.4.5] 5. Lines 119-128, the differentiation time for Orchidaceae is inconsistent with that in the published orchid genome papers. Please provide an explanation.

Response:

*It is correct that our estimates for ages of/within the orchids are comparatively younger than what is stated in some of the orchid genome papers. As a basis for our age estimation, we used the time of origin of orchids as estimated in Kim et al. (2020), that is 99.20 (83.78–114.92) Ma. Kim et al. (2020) conducted divergence estimation based on a phylogeny of 124 orchid species using BEAST2. The phylogeny of 124 species more accurately reflects the evolutionary history of orchids as compared to genome papers considering only few orchid species (e.g., Zhang et al., 2017; Li et al., 2022). Therefore, we considered Kim et al.'s (2020) values of 83.78 Ma and 114.92 Ma as the minimum- and maximum-age calibrations of the stem age of orchids. We inferred the origin time of Orchidaceae and Orchidoideae to be 99 Ma and 54.49 Ma, respectively, which closely align with the estimates of 99.96 Ma and 59.16 Ma by Kim et al. (2020). In addition, we note that the maximum age we used to calibrate the stem age of orchids, at 114.92 Ma is not significantly different from, e.g., the estimates of 128 Ma in Zhang et al. (2017) or 125 Ma in Li et al. (2022). Nonetheless, given our age calibrations our inferred divergence times are younger when compared to several recent orchid genome papers (although there are also counter-examples, e.g., the *Cymbidium goeringii* paper by Chung et al. 2021 provides an estimate of 69 Ma for the split of *Apostasia*, which is slightly younger than our estimated of 71 Ma). In order to acknowledge this, we have added a sentence to the manuscript:*

[line 150/151] “ Our age estimates for orchids, while in line with results by Kim et al. (2020)⁵⁰, are younger when compared with previous orchid genome studies^{28,31}. ”

Reviewer #2

[R2.1] This exciting and highly readable manuscript reports the 5.2 GB chromosome-scale genome sequence of the sexually deceptive orchid, *Ophrys sphegodes*. While genome sequencing, assembly and annotation is increasingly feasible, an effective team of multi-disciplinary specialists is still required. It is clearly evident that such an expert team has been involved in this project.

Collectively this outstanding team have achieved several key firsts. For example, from a genome perspective, this team has successfully tackled the largest orchid genome to be sequenced to-date. Further, while it is not surprising that such a large orchid genome will be characterised by the highest abundance of long terminal repeat elements (relative to other sequenced orchid genomes), this study has taken on the challenge of building the first species level database of orchid transposable elements. Such a database promises to be a powerful resource for many future studies. The study also anchored the new orchid genome with a phylogenomic analysis, divergence time estimation and analysis of genome size expansions and contractions across a total of 21 orchid and outgroup genomes. Finally, this is also the first orchid genome to be sequenced, where the species is obligately dependent on a single pollinator species, in the highly specialised pollination strategy of sexual deception, offering an exemplar case study where pollinator-driven evolution is likely to leave a strong genomic signature.

The strategically chosen study species, *Ophrys sphegodes*, has been the subject of prior in-depth ecological, evolutionary, chemical and molecular studies, while the genus at large is one of the most intensively studied of all wild plant genera. Building on this exceptionally rich background knowledge, this study has gone beyond other genome papers to draw strong connections and links between this background knowledge, and new findings from the genome of a wild species. As one example, the study included detailed investigations of three gene families predicted to be closely linked to the production of specific alkenes, as key components of the sex pheromone mimicry which underpins pollinator attraction. Relative to the closely related *Platanthera* orchid genome, the findings confirm extensive gene duplication, with such novel genes plausible candidates for pollinator driven

selection. Without the extensive prior work on the chemistry and biosynthesis of sex pheromone mimicry, these key insights from genome are unlikely to have been found. Finally, the outcomes of an innovative population genetic study of *O. sphegodes* and three closely related sister taxa is reported. Despite, extensive genome wide similarity and high levels of allele sharing, a 20 mb ‘genomic island of speciation’ has been found between *O. sphegodes* and its sympatric sister *O. exaltata*, which warrants more detailed investigation in the future.

In conclusion, this study highlights the value of building genomic resources to complement even a well-studied wild species. Furthermore, as the largest orchid genome to be sequenced to-date, the study demonstrates the potential to now tackle the genome sequencing of many other orchids of this genome size and larger.

Response:

Thank you for your positive assessment. We share the hope that this paper will ultimately help in understanding the strategy of sexual deception more broadly, which is now also reflected in the concluding sentence of the abstract.

We agree that the strikingly differentiated ‘island’ on chromosome 2 warrants further investigation. We have now, in response to comment R1.2.4 by Reviewer 1, included several analyses in the manuscript (please see our response to that comment for details), but it is also clear that this is only the first step and that future studies dedicated to this point will have to be conducted.

[R2.2] Despite my great enthusiasm for this study, I was surprised by the claim on line 142 of ‘the unique pollination system’. To the contrary, sexual deception is a worldwide phenomenon with multiple evolutionary origins across the Orchidaceae. Indeed, hundreds of orchid cases are known from across four continents (see Current Biology 33 2023 R453-R518 for a recent plain English review), with species radiations potentially even larger than found in *Ophrys*. Intriguingly, only two cases of sexual deception are known from outside the orchid family, begging the question of why this strategy is far more prevalent in the orchids. Building on this first genome case study, high quality genomic resources across a diversity of sexually deceptive orchids could help unlock the key to this and many other seemingly mysterious aspects of sexual deception.

I can only assume this claim of ‘unique pollination’ was an unintentional oversight, which can be easily rectified with a small expansion of the text, and one or two additional supporting references. Furthermore, there is a missed opportunity to set up for future studies some genomic hypotheses that might apply to the repeated evolution of sexual deception and the very frequent evolution of specialised pollination in the orchids more generally.

Response:

*Indeed, we meant the ‘unique pollination system’ of sexual deception rather than of *Ophrys*, but we acknowledge that the word ‘unique’ is not helpful here. We have therefore removed it and simply refer to ‘sexual deception’. Additionally, in line with this comment, comment R3.3 by Reviewer 3 and several remarks by Reviewer 1 regarding sexual deception, we have expanded the section by inserting additional text at the beginning of the paragraph (also citing the plain-English review mentioned).*

The original section start was

“To gain insights into the genomic basis of the unique pollination system, we ... ”

The revised text [lines 165ff] now reads...

*“Sexual deception is not restricted to the Euro-Mediterranean genus *Ophrys*, but is a worldwide phenomenon. It originated several times independently and there are many examples among Australian orchids, whereas only a few non-orchid cases, such as the South African daisy *Gorteria diffusa*⁵⁵, are known. Although specific pollinator interactions mediated by floral chemistry are a common theme in sexual deception^{53,54}, it remains unknown why this pollination strategy occurs predominantly among orchids and what allowed its repeated*

evolution in this family. To gain insights into the genomic basis of sexual deception in *Ophrys*, we ...”

[R2.3]

Below I offer some additional comments, suggestions and queries.

[R2.3.1] Ln 55, 60. In order to provide a balanced perspective, I suggest a short sentence with supporting reference(s) that notes that there is some literature challenging the high degree of specificity and the number of *Ophrys* species. Perhaps something like: “Notwithstanding some uncertainty about the number of species and the extent of pollinator sharing, at the local population level extreme specialisation is evident in most *Ophrys* species”.

Response:

This is a good point. We have added the sentence (line 66ff) as suggested by Reviewer 2, adding references to Vereecken et al. (2011) and Bateman et al. (2011) who discuss this issue at length.

[R2.3.2] Ln 86. Suggest or expanding sentence here as in the Supplement to include ‘function, gene ontology (GO), and protein domain ...’

Response:

Following this comment, after the statement

[now line 100f] “..., of which 90.01% had functional annotation or RNA-seq support (see Supplementary Tables 7-10 for genes of interest) ”

we added: “, which included information on Gene Ontology (GO) terms, protein domain information, putative pathways and enzyme function.”

[R2.3.3] Ln 94, ‘novel sequences’ or ‘novel sequence motifs’?

Response:

These are full sequences with a considerable length – certainly longer than what we would consider a ‘motif’. We therefore think that the original wording is appropriate and kept it.

[R2.3.4] Ln 104, suggest it would be helpful to add a short plain English sentence on how this age estimation was done. Alt. give the assumptions used to make the estimates.

Response:

Thank you for this good idea. Given that technical details are given elsewhere in the manuscript, we inserted the following non-technical explanation in line [now] 119ff:

“We conducted an analysis of LTR insertion age based on the idea that both LTR sequences of a TE are identical at the time of insertion, but will diverge over time as mutations accumulate. ”

[R2.3.5] Ln 185, suggest it would be helpful to annotate SAD gene numbers around the circle of Fig 3, for reader ease. Otherwise, the reader is forced to infer location from the terminal labels, which are quite hard to read.

Response:

Thank you for this suggestion regarding what is now Fig. 4. We tried this but found the figure became too crowded when annotating them all in bigger size. Instead, we highlight the two main clades of interest, the SAD2 clade (SAD1/2/7/8) and the SAD5 clade (SAD5/6/9/10) with curly braces.

[R2.3.5] Ln 200, is '15 gene models' correct? Suggest, '15 of the 16 FAR homologues form a phylogenetic clade ...'

Response:

Yes, technically '15 gene models' is correct, but in this case identical with '15 homologues'. Since we agree that the suggested wording (with 'homologue') is clearer, we have changed the sentence accordingly. The revised sentence reads:

*"Fifteen out of the 16 FAR homologues found in the *O. sphegodes* genome form a phylogenetic Orchidoideae clade together with a single *Platanthera* sequence."*

[R2.3.6] Ln 221, suggest replacing 'majorly' with 'make a major'.

Response:

We have changed this accordingly.

[R2.3.7] Ln 267, Suggest it would be helpful to the reader to spell out the final number of individuals used for sequencing.

Response:

The genome assembly was done from a single individual (data from a second individual were later round in the third polishing step, see SI; but these reads were not used for assembly). Since this was not sufficiently clear in the preceding sentence, we inserted the word 'single' so that it now reads:

*"The single *O. sphegodes* individual for the genome assembly (accession SPH_8) was selected among several samples previously collected ..."*

Likewise, we clarified in line 340: "DNA isolated from accession SPH_8 was used to prepare ..." We hope this is sufficiently clear now.

[R2.3.8] Finally, based on the mss alone, I was confused as to when Illumina sequencing was used. If I understand correctly from the supplement, Illumina sequencing was used to generate additional short-read libraries, and for the Hi-C. If so, I suggest minor changes in the main text to clarify:

Ln 73/74. Suggest adding 'sequenced with Illumina'

Response:

Thank you for pointing this out. We have clarified this now by changing the sentence:

[old] "To assemble the genome, we generated a total of 409 Gb data on the Nanopore PromethION platform (Supplementary Table 2, Supplementary Fig. 3) and sequenced Hi-C chromatin conformation capture libraries (Supplementary Table 3, Supplementary Fig. 4)."

to

[revised, lines 82ff] "To assemble the genome, we generated a total of 409 Gb data on the Nanopore PromethION platform (Supplementary Table 2, Supplementary Fig. 3). Additionally, whole-genome Illumina sequencing data (WGS; 268 Gb, Supplementary Table 3) and Hi-C chromatin conformation capture libraries (Illumina, Supplementary Table 3) were produced to perform polishing and anchoring of scaffolds, respectively."

[R2.3.9] Ln 81/82. Suggest adding "Hi-C" Illumina sequence reads.

Response:

It should be "WGS" rather than "Hi-C". We have added "WGS" so as to clarify this in line 93.

[R2.3.10] Ln 284-292. Please spell out use of Illumina sequencing more clearly.

Response:

Indeed, we only mentioned Illumina data later in the paragraph. We therefore clarified this and inserted mention of ‘two Illumina’ libraries in the first sentence, which now reads [line 340]: “Isolated DNA isolated from accession SPH_8 was used to prepare two Illumina and eight ONT libraries ...”

This also better clarifies that a single sample was used (cf. query R2.3.7)

Rod Peakall
The Australian National University

Reviewer #3

Dear Authors:

[R3.1] I enjoyed reviewing your paper “The genome of the early spider-orchid *Ophrys sphegodes* provides insights into sexual deception and adaptation to pollinators”. I found it to be clearly written and interesting.

I believe this is a valuable paper for the pollination field, especially for systems relying on sexual deception which has not received as much attention in terms of their genetic architecture as other pollination systems. Having a genome of *Ophrys* available and identifying the likely means by which it has adapted and radiated over a short period of time is a breakthrough.

I found the methods you provided clear and comprehensive. The results are sensible and clearly interpreted, which will make this paper valuable to anyone interested in sexual deception. It was interesting to note the dominance of repetitive elements and the role they have played, and it would be great to see if this matches radiations in other sexually deceptive plants.

Response:

Thank you for your feedback and positive assessment of our work.

[R3.2] The figures are mostly clear, although I would suggest a few amendments. You might consider removing the number of expanded and contracted gene families on the branches in Fig. 2a, as the bubbles at the tips clearly depict this information and will make for a cleaner image. It would also be useful to add a thumbnail photo of a flower of *O. sphegodes* and *P. zijinensis* in Fig. 2b (as you have done in Fig 4b). For Fig. 4, naming each species and their abbreviations/colours at the start in Fig. 4a would improve clarity as this information currently only appears at the end of the legend. It may be even better to include it in the figure itself, as readers will likely view this before reading the full legend.

Response:

Thank you for these good suggestions. We have accordingly made several small changes to the figures:

- *Fig. 2. In panel 2b, we included small thumbnails for *Ophrys* and *Platanthera* (although, for lack of a photo, a different species) in the tree and we agree that this communicates well. Regarding panel 2a, we agree that removing the numbers would make it prettier; however, replacing them with bubbles did not seem to make matters better. Also, in terms of interpretation, the expansions/contractions at branch points are always relative to the higher-order branches, so we found that, on balance, it was better for the numbers to remain there.*

- *Fig. 5 (previously Fig. 4): As suggested, we include the species labelling in full inside the figure. We agree that this is easier for the reader.*

[R3.3] To enhance the range of this paper, it would be great to include a bit more context about sexual deception and where it occurs. I realise there are space limitations, but some mention here of other sexually deceptive systems outside of Europe that have also undergone recent diversification in response to becoming sexually deceptive would be informative. Adding a more global context of this pollination strategy will position this paper better within the field, especially considering your focus on sexual deception.

Response:

Thank you for this suggestion, which echoes comment R2.2 (and also R1.4.2 and several of Reviewer 1's points). We agree that this is a useful addition and have added relevant text. Please see comment R2.2 for the detailed changes.

[R3.4] A stronger focus on pollinators, in particular the behaviour of deceived pollinators and the selection they exert on sexually deceptive orchids, would help support the conclusions and claims made here, especially considering your use of "adaptation to pollinators" in the title. Adding these angles will also help to highlight the novelty of this paper.

Response:

This echoes Reviewer comments R2.2 and R1.4.2. We agree that in our original manuscript the mention of these points may have been too brief, primarily relying on citations to other papers. We agree with these comments and have therefore added text to introduce key aspects of Ophrys floral adaptation to pollinators and think that this indeed benefits the paper. In an effort to remain as concise as possible, we have focused on findings from the Ophrys sphegodes s.l. lineage (save for the paper by Stejskal et al. 2015), as this is the focus of the current manuscript.

In line 54, we have introduced the following text:

"Among the flower traits adapted to pollinators, olfactory signals are pivotal to specific pollinator attraction^{10,11}, with selection by pollinators¹² leading to strong odour differentiation among closely related species¹³. Additional adaptations to pollinators likely involve flower labellum colour, which matches pollinator body coloration, and floral morphology that optimises pollen transfer¹³. At the same time, conspicuous UV-reflective patterns and odour compounds not primarily required for sexual attraction are highly variable between plants and likely aid male pollinators in memorising and avoiding plants, thereby increasing outcrossing rates¹³⁻¹⁵."

We realise that this statement still does not discuss the full list of adaptations to pollinators that have been proposed to exist in Ophrys flowers. However, we think that a more in-depth discussion of this topic is beyond the scope of this paper.

[R3.5] I appreciate the inclusion of a phylogenomic tree comprising several angiosperms for which full genomes are available. This is a welcome addition and should also be of interest to readers beyond those working on sexual deception.

Response:

Thank you for this comment echoing comment R2.2. Please note that in response to query R1.4.5, we have slightly amended the discussion of this analysis to mention our younger age estimate for orchid groups.

Other changes to the manuscript

Several additional changes to the manuscript have been carried out to keep the manuscript readable, adhere to journal style and to correct minor errors.

- *Several small changes were carried out to conform to the Nature Communications manuscript guidelines:*
 - *the title was reworded to have no more than 15 words*
 - *the abstract was shortened to below 150 words*
 - *some section headings were renamed*
 - *the previously separate funding information was merged into the Acknowledgements section*
 - *a Competing Interests statement was added (the authors have no competing interests)*
- *Several small editorial changes to improve the clarity of wording throughout the text*
- *[lines 429 and 448] The version numbers of MUSCLE and BWA-MEM2 were incorrectly stated and have been corrected (MUSCLE v5.1.0; BWA-MEM2 2.2.1)*
- *In the author contributions section [line 738] the word “collected” Ophrys samples was incorrect and was replaced by “provided”. The work of M.A. for the revision of the paper was added as his contribution to gene expression and GO analysis.*
- *Acknowledgements – we added thanks to two students who had supported us during some analyses.*
- *Fig. S1 – flow cytometry figure has been updated to conform with Nature’s reporting guidelines for flow cytometry data.*
- *Minor corrections to Table S8 have been carried out: corrected spelling (anthocyanidin vs anthocyanin), updated information on AGXT1 (spelled AGT1 in the original version); Anthocyanidin 3-O-glucosyltransferase was incorrectly referred to as UDP-glucose:flavonol 3-O-glucosyltransferase in the original manuscript.*
- *Data availability section has been updated to include new accession numbers and the Figshare DOI.*

Reviewers' Comments:

Reviewer #1:

Remarks to the Author:

In general, the analyses and manuscript are much improved. Many of my comments have been addressed. However, there are still defects that I would like to point out. First, the RNA-seqs used in this study were derived from previous published dataset, although the authors claimed that they did the differential gene expression analysis. In addition, the RNA-seqs just came from mature labellum of *Ophrys sphegodes* and *O. exaltata*, respectively. From the developmental point of view, it is not of great significance for understanding the specificity of labellum development for comparing differentially expressed genes (DEGs) between *O. sphegodes* and *O. exaltata*, especially since the two species are related species within the same genus. I suggest that the authors should prepare each of floral organs and several distinct floral developmental stage tissues, then compare the DEGs among these samples. Thus, the conclusion will be more convincing. Similar concerns also are considered for hydrocarbon biosynthesis and anthocyanin biosynthesis.

Second, For the [R1.2.4], I am impressed that authors adopted different strategies to screen 7 genes probably involved in differentiating *O. exaltata* from *O. sphegodes*. I suggest the expression patterns of the 7 genes and their annotation should be included in the supplementary figure 18. If the authors could include the spatial and temporal expression patterns from RNA-seqs dataset, the results will be better. In addition, the authors should provide additional explanations for how these genes' functions are contributing to pollination isolation, not just regulatory rather than biosynthetic genes.

Minor

1. In Supplementary Figure 10, why the genes in AGL12 group are separated into different clades?
2. According to the phylogenetic tree in Supplementary Figure 10, I recognized there are at least 6 genes belonging to SEP genes in *O. sphegodes*. However, the authors just list the expression patterns of 4 genes of *O. sphegodes*.
3. In the Supplementary Figure 11, AGL7 should be corrected.
4. I do not think the number of SEP genes in *Platanthera zijinensis* is correct.
5. Several figures showed typo of *zijinensis*, such as Figure 2. Please check all the contents.
6. In Supplementary 13, I do not agree that the number of genes involved in anthocyanin biosynthesis is correct in the genome sequenced orchids. For example, the DFR and ANS in *Phalaenopsis equestris* should exist in the genome, because expression of DFR and ANS have been examined, and the species contains anthocyanin (Hsu et al., 2015). The authors should mention how they identified the genes involved in anthocyanin biosynthesis.
7. Again, in Figure 3, Why the authors did not identify KAR in the genome sequenced orchids except the species in Orchidoideae. Please mention how did you identify the genes.

Reviewer #2:

Remarks to the Author:

The authors have carefully responded to the minor concerns and suggestions I raised in my earlier review of this manuscript. Importantly, they have now clearly identified that pollination by sexual deception is a world-wide phenomenon, with multiple independent origins. Furthermore, given this is the first genome to be published for any sexually deceptive species, the authors have taken the opportunity to highlight the potential value of this genome study for future investigations into the repeated evolution of sexual deception. Thus, rather than the previous focus on a 'unique pollination strategy', the wider significance is now clearly articulated.

It follows, that this exciting and highly readable study will attract citations from a wide range of studies well into the future. I predict this will include, but not be limited to, other orchid genome

studies, genome studies of pollinator-mediated evolution, genome studies into repeated evolution of highly specialised pollination strategies, such as sexual deception, genome studies of other sexually deceptive plants etc. I also predict that the evidence for a key role of gene duplication in pollinator adaptation, the strong evidence for genic evolution, and the very high incidence of TEs uncovered in this genome will all be topics of wide interest. Finally, it is worth highlighting that as the first genome of a sexually deceptive orchid, and in indeed the first genome in a diverse group of European terrestrial orchids, this study provides a crucial foundation that will underpin many additional studies far into the future.

Beyond their effective response to my suggestions (as Reviewer #2), the authors have provided a very extensive and carefully argued point-by-point response to Reviewer #1, whom while enthusiastic about the study, raised the longest list of queries and suggestions.

I concur with the authors that it is not yet feasible to address all of the interesting additional analyses suggested by the Reviewer #1. In particular, with orchid genomes available for less than 10 independent genera, and given the sheer diversity of the Orchidaceae with more than 25,000 species across 100's of genera, there remains much unknown about the genomic basis of orchid evolution. In fact, this is all the more reason why this present study is of critical importance. Not only is it the first genome of an orchid with a highly specialised pollination system, but it is also the first genome in a diverse branch of the important terrestrial orchid subfamily the Orchidoideae. Additional genomes of related and non-sexually deceptive genera will be essential to fully unravel the evolution of sexual deception in Ophrys. However, until they become available it is crucial to be avoid over interpretation of the Ophrys sphegodes genome.

Nonetheless, in this substantial revision of the manuscript, the authors have considerably expanded the number of analyses, and cautiously interpreted the findings to give expanded insights and new hypotheses for detailed future exploration. These additional analyses include: 1. Expanded comparative analysis of hydrocarbon, anthocyanin and MADS-box genes, which provide strong evidence for several relevant gene family expansions that are plausibly linked to the evolution of sexual deception. However, genomes of non-sexually deceptive are required to pin down the timing and evolutionary role of these expansions. 2. Maps of the plausible biosynthetic pathways of the hydrocarbons and anthocyanin are also shown, along with supporting gene expression results. New evidence for differential expression at some relevant hydrocarbon genes will warrant more detail investigations in the future. 3. New Fst analysis based on RNA-sequence data between *O. sphegodes* and closely related *O. exaltata* which largely reinforces earlier GBS results of a chromosome 2 species level 'island of differentiation'. 4. Addition of GO enrichment analysis of this region of differentiation. Intriguingly, the overall findings from the expanded analysis of the 'island of differentiation' point to regulatory genes rather than hydrocarbon or anthocyanin genes in this region. If true, it may indicate an overlooked role of changes in regulation as a major contributor to genic speciation. Clearly, this possibility warrants detail investigation in future studies.

Minor suggestions.

Ln 110-118

The comparisons of estimated percentages of LTR elements are reported to 2 decimal places. However, given that any published genome is never 100% complete, and given some uncertainty in the annotation pipelines, I suggest that it is will never be possible to estimate the percentage of LTR elements to this 2 decimal place level of precision. Surely an integer estimate is even optimistic?

Elsewhere, I note some 2 decimal place estimates, e.g. Line 100 with 90.01% etc, which should be simplified to integer or at most 1 decimal place.

Ln 166. Insert 'has' i.e. It has originated several times independently ...

Rod Peakall
The Australian National University

Reviewer #3:

Remarks to the Author:

I have now reviewed the revised manuscript "Genome of the early spider-orchid *Ophrys sphegodes* provides insights into sexual deception and pollinator adaptation".

The authors have done a good job of addressing my concerns, as well as that of the other reviewers, which makes for a more comprehensive and balanced paper.

The images are informative and clear, especially the inclusion of the new figure on the hydrocarbon biosynthesis pathway (Figure 3).

The larger field of sexual deception is also acknowledged and briefly discussed, including how sexually deceptive flowers function and are pollinated. Inclusion of more ecological discussion would have been preferable, but as this is primarily a genomic study. I think this is fair.

The authors have contributed significant data that will hopefully open the door to more comparative analyses as more genomes of orchids with different pollination strategies become available.

I think this paper will be of interest to readers and is a valuable contribution to the field.

RESPONSE TO REVIEWER COMMENTS

Reviewer #1 (Remarks to the Author):

In general, the analyses and manuscript are much improved. Many of my comments have been addressed. However, there are still defects that I would like to point out. First, the RNA-seqs used in this study were derived from previous published dataset, although the authors claimed that they did the differential gene expression analysis. In addition, the RNA-seqs just came from mature labellum of *Ophrys sphegodes* and *O. exaltata*, respectively. From the developmental point of view, it is not of great significance for understanding the specificity of labellum development for comparing differentially expressed genes (DEGs) between *O. sphegodes* and *O. exaltata*, especially since the two species are related species within the same genus. I suggest that the authors should prepare each of floral organs and several distinct floral developmental stage tissues, then compare the DEGs among these samples. Thus, the conclusion will be more convincing. Similar concerns also are considered for hydrocarbon biosynthesis and anthocyanin biosynthesis.

Response:

Thank you for your feedback. There are a few points to make about the RNA-seq analysis.

Firstly, practically speaking (and as outlined in our responses to the first round of reviews to this manuscript), we cannot provide such an RNA-seq data set at present, our only source for relevant plant material being from natural populations. While we attempted to collect material for expression analysis in March 2024, the extreme temperatures had shifted flowering time forward so that we arrived too late to sample bud material (or even good fresh flowers). Therefore, our next chance to collect material would be March/April 2025 and any RNA-seq analysis would unlikely to be complete before Q3/Q4 2025. We do believe – and we hope Reviewer 1 will agree – that we should not delay the publication of the *Ophrys* reference genome (and any other manuscripts depending on it) for so long in order to add a data set that, as we shall argue below, is not at the heart of this study.

Secondly, we can indeed confirm that we did carry out the DE analysis of this previous dataset, which was generated in the senior author's lab and was from genetic material of the same species as the rest of this manuscript. That data set was originally intended for performing DE analysis between the species of the Italian *O. sphegodes* group, yet due to reasons unconnected with science, this had never actually happened prior to the present analysis. We can therefore confirm that the analyses shown were entirely novel.

Finally, we agree that this data set from mature labella is not ideal to study floral development. However, we would like to point out that this had not been the main aim of the present study, the key focus of which is on providing a genome reference so as to better understand species divergence. Therefore, we respectfully disagree with Reviewer 1 about the necessity to include such a data set in the present manuscript (incidentally also echoing comments by Reviewer 2). Having said this, we do agree that it would be interesting to generate the data set Reviewer 1 asks for in the course of a future project. As regards anthocyanin and especially alkene synthesis, our previous published (Schlüter *et al.* 2011 *PNAS*; Xu *et al.* 2012 *PLoS Genet.*; Sedeek *et al.* 2016 *Curr. Biol.*) and unpublished data suggest that biosynthesis and phenotype formation occur late in development and peaks at anthesis, so that data on earlier developmental steps are probably less important here.

Second, For the [R1.2.4], I am impressed that authors adopted different strategies to screen 7 genes probably involved in differentiating *O. exaltata* from *O. sphegodes*. I suggest the expression patterns of the 7 genes and their annotation should be included in the supplementary figure 18. If the authors could include the spatial and temporal expression patterns from RNA-seqs dataset, the results will be better. In addition, the authors should provide additional explanations for how these genes' functions are contributing to pollination isolation, not just regulatory rather than biosynthetic genes.

Response:

*Thank you for this good suggestion. We have revised Suppl. Fig. 18 accordingly. The figure now shows gene expression information. Since it was not possible to squeeze the annotation information into the figure as well, the figure legend now refers to a new table, which we have included to provide

this information (new Supplementary Table 21). As regards further insights into how these genes affect pollinator-mediated reproductive isolation, we regret to say that currently, we simply do not know. It is something we will have to address in future studies, but at this stage we do not have any useful speculation to add to the manuscript. However, like Reviewers 2 and 3, we too expect that several future studies will have to follow up on some of the questions raised by the present study.

Minor

1. In Supplementary Figure 10, why the genes in AGL12 group are separated into different clades?

Response:

In Suppl. Fig. 10, we chose to visualise only the *Ophrys* MADS box genes for the simple reason that (i) *Ophrys* is the focal organism of this study and (ii) we found it visually overwhelming to include additional organisms. The phylogenetic tree depicted is based on *Ophrys* sequences, using the entire annotated gene length. In contrast, our assignment to MADS clades is based upon a larger data set and the comparison with genes and data from other organisms.

In the case of *AGL12*, there are three *Ophrys* copies: Osph3G80660.1, Osph19G17210.1 and Osph19G17430.1. The latter two are truncated, as the below snippet out of a combined phylogenetic and protein motif discovery analysis via MEME shows:

(We note that there is no *a priori* reason to discard truncated MADS proteins in our annotation, since they may well be functional, e.g. Liu *et al.* 2020 *J. Exp. Bot.*;doi: 10.1093/jxb/eraa116)

We suspect that because there is no sensible alignment possible for the 3' portion of the gene, the phylogenetic analysis used for Suppl.Fig.10 separates the truncated from the non-truncated forms (note that the position of Osph3G80660.1 has little bootstrap support), although all three *Ophrys* genes cluster with *bona fide* *AGL12* genes such as *Arabidopsis AGL12* (AT1G71692), when included in a joint analysis (here a snippet of a tree with rice and *Arabidopsis*):

While it would be nicer to recapitulate this visually in Suppl.Fig.10, on balance we still think the tree we show is the best concise summary of *Ophrys* MADS-box genes we can provide. Importantly, considering the total evidence available, we consider our MADS subgroup assignment to be accurate and useful for future users of the *Ophrys* genome annotation.

2. According to the phylogenetic tree in Supplementary Figure 10, I recognized there are at least 6 genes belonging to SEP genes in *O. sphegodes*. However, the authors just list the expression patterns of 4 genes of *O. sphegodes*.

Response:

In the previous manuscript version, Suppl. Fig. 10 incorrectly showed two *AGL6* genes coloured as *AGL2*, giving an incorrect idea about the number of *SEP* (= *AGL2/3/4/9*) genes. The number of 4 *O. sphegodes* *SEP* genes appears to be correct, however. The figure in question has hence been corrected. Note that, as with the previous comment by Reviewer 1, we have double checked and consider the *Ophrys* *AGL2* and *AGL9* gene assignments to be correct.

3. In the Supplementary Figure 11, *AGL7* should be corrected.

Response: Thank you for spotting this typo. It should have been “*AGL2*” and we have corrected it accordingly.

4. I do not think the number of *SEP* genes in *Platanthera zijinensis* is correct.

Response:

We agree that, biologically speaking, homologues for all the MADS-box gene classes depicted in Supp. Fig. 11 are expected to exist in all shown orchids. Therefore, we concur that we would have expected to identify *SEP* genes in *P. zijinensis*.

In response to this point, we investigated why we did not find any *Pzi* *SEP* genes; we repeated parts of our analysis and pursued alternative analyses (now also adding *Cymbidium* to our analysis). In short, we only have access to the coding sequences (CDS) but not the full annotation data for all genomes and can therefore only assign putative homologues based on sequence similarity to genes we have ourselves functionally annotated. To do so, for each gene class of interest, we identified all orthogroups containing *bona fide* representative sequences, e.g. from *Arabidopsis*, or genes from our own *Ophrys* functional annotation. We then counted orthogroup members for each orchid genome of interest. We acknowledge that this approach has limitations and that it may not always be possible to recover all homologues that are actually present in the genomes (e.g. we cannot identify a homologue missing from the CDS files we obtained).

In the case of orchid *SEP* (specifically, *AGL2* and *AGL9*) and *AGL6* sequences, sequences were split over three orthogroups, yet the orthogroup boundaries did not coincide exactly with what we considered *AGL2*, *AGL9* and *AGL6*-like sequences. In addressing this point, we therefore pooled sequences from all three orthogroups (including *AGL* sequences which were annotated as, among others, *AGL16*, *AGL44*, *AGL21*, *AGL17* in *A. thaliana* and *O. sativa*) and re-analysed them phylogenetically (both at amino acid and nucleotide level); the analysis included sequences from the annotated orchids *A. shenzhenica* and *P. equestris* as well as *A. thaliana* and *O. sativa*. We then proceeded to look in detail at the MEME motifs of the known *SEP* sequences within our orthogroup assignment. However, after we delimited the genes included in the *SEP* from the *AGL6* clade for all the orchids in this study, the assignment of some of the shorter (truncated) orchid sequences to *AGL* subclades remained unclear. Specifically, with several sequences from *P. zijinensis*, we cannot confirm that they definitely belong to the *SEP* clade. These ambiguous sequences are therefore not counted and hence not included in our Suppl. Fig. 11. To clarify our use of orthogroup data to estimate gene copy number, in the respective figure legend we specified:

“Gene copy numbers, estimated from orthogroup membership, ..” (newly inserted text underlined).

5. Several figures showed typo of *zijinensis*, such as Figure 2. Please check all the contents.

Response: Thank you for spotting this. We have gone through the entire text, SI, figures and tables and fixed this in several places, i.e., in Fig. 2b, Suppl. Fig. 9b, Suppl. Table 18 and main text line 438.

6. In Supplementary 13, I do not agree that the number of genes involved in anthocyanin biosynthesis is correct in the genome sequenced orchids. For example, the *DFR* and *ANS* in *Phalaenopsis equestris* should exist in the genome, because expression of *DFR* and *ANS* have been examined, and the species contains anthocyanin (Hsu et al., 2015). The authors should mention how they identified the genes involved in anthocyanin biosynthesis.

Response:

As with the previous point about *SEP*, we agree with Reviewer 1 that we would have expected to find *DFR* and *ANS* in *Phalaenopsis*. Gene class assignment was done from orthogroup data, similarly as with the MADS-box genes, with the difference that for biosynthetic genes our genome annotation also contains information on pathway (KO) and enzyme (EC) information that we draw upon. However, in principle, the same limitations apply as outlined in our response to point 4.

In response to this comment, we repeated the analysis and double checked the numbers in the figure. It turns out that the numbers were correct, but in preparing the figure, we had swapped the columns for *Phalaenopsis* and *Gastrodia*. Since *G. elata* is saprophytic, has white, reduced flowers and has suffered extensive gene deletion (Yuan et al. 2018 *Nat. Commun.*), it is conceivable that it actually lacks *ANS* and *DFR*. At the same time, we can confirm that *Phalaenopsis* does contain *ANS* and *DFR*. We have fixed the error in the figure.

To clarify our approach, we inserted the following sentence into the figure legend:

“Gene copy numbers were estimated by tallying the orchid members of orthogroups containing functionally annotated anthocyanin biosynthetic genes.”

7. Again, in Figure 3, Why the authors did not identify *KAR* in the genome sequenced orchids except the species in Orchidoideae. Please mention how did you identify the genes.

Response:

Thank you for pointing this out. We re-checked the analysis and the same caveats as with the previous two points apply. While we therefore cannot guarantee to have found every homologue, in the case of hydrocarbon biosynthetic genes, we realised that additional orthogroups had to be considered, which contained *Arabidopsis* (but no *Ophrys*) biosynthetic genes. This affected *KAS*, *KAR*, *LACS* and *KCS* genes. We can thus confirm that all sequenced orchid genomes contain *KAR* copies. We note that several biosynthetic genes are missing for *Cymbidium* – potentially consistent with the large number of gene family contractions (Fig. 2) for this plant. We have been unable to find respective homologues in the *Cymbidium* data, even though we would expect this plant to contain core very-long-chain fatty acid biosynthetic genes; we therefore cannot exclude the possibility that this is a technical issue associated with the *Cymbidium* genome data.

Figure 3 was updated accordingly (and also a typo fixed). Analogously to point 6, we inserted the following sentence into the figure legend:

“Gene copy numbers were estimated by tallying the orchid members of orthogroups containing functionally annotated hydrocarbon biosynthetic genes.”

Reviewer #2 (Remarks to the Author):

The authors have carefully responded to the minor concerns and suggestions I raised in my earlier review of this manuscript. Importantly, they have now clearly identified that pollination by sexual deception is a world-wide phenomenon, with multiple independent origins. Furthermore, given this is the first genome to be published for any sexually deceptive species, the authors have taken the opportunity to highlight the potential value of this genome study for future investigations into the repeated evolution of sexual deception. Thus, rather than the previous focus on a ‘unique pollination strategy’, the wider significance is now clearly articulated.

It follows, that this exciting and highly readable study will attract citations from a wide range of studies well into the future. I predict this will include, but not be limited to, other orchid genome studies, genome studies of pollinator-mediated evolution, genome studies into repeated evolution of highly specialised pollination strategies, such as sexual deception, genome studies of other sexually deceptive plants etc. I also predict that the evidence for a key role of gene duplication in pollinator adaptation, the strong evidence for genic evolution, and the very high incidence of TEs uncovered in this genome will all be topics of wide interest. Finally, it is worth highlighting that as the first genome of a sexually deceptive orchid, and in indeed the first genome in a diverse group of European terrestrial orchids, this study provides a crucial foundation that will underpin many additional studies

far into the future.

Beyond their effective response to my suggestions (as Reviewer #2), the authors have provided a very extensive and carefully argued point-by-point response to Reviewer #1, whom while enthusiastic about the study, raised the longest list of queries and suggestions.

I concur with the authors that it is not yet feasible to address all of the interesting additional analyses suggested by the Reviewer #1. In particular, with orchid genomes available for less than 10 independent genera, and given the sheer diversity of the Orchidaceae with more than 25,000 species across 100's of genera, there remains much unknown about the genomic basis of orchid evolution. In fact, this is all the more reason why this present study is of critical importance. Not only is it the first genome of an orchid with a highly specialised pollination system, but it is also the first genome in a diverse branch of the important terrestrial orchid subfamily the Orchidoideae. Additional genomes of related and non-sexually deceptive genera will be essential to fully unravel the evolution of sexual deception in *Ophrys*. However, until they become available it is crucial to be avoid over interpretation of the *Ophrys sphegodes* genome.

Nonetheless, in this substantial revision of the manuscript, the authors have considerably expanded the number of analyses, and cautiously interpreted the findings to give expanded insights and new hypotheses for detailed future exploration. These additional analyses include: 1. Expanded comparative analysis of hydrocarbon, anthocyanin and MADS-box genes, which provide strong evidence for several relevant gene family expansions that are plausibly linked to the evolution of sexual deception. However, genomes of non-sexually deceptive are required to pin down the timing and evolutionary role of these expansions. 2. Maps of the plausible biosynthetic pathways of the hydrocarbons and anthocyanin are also shown, along with supporting gene expression results. New evidence for differential expression at some relevant hydrocarbon genes will warrant more detail investigations in the future. 3. New Fst analysis based on RNA-sequence data between *O. sphegodes* and closely related *O. exaltata* which largely reinforces earlier GBS results of a chromosome 2 species level 'island of differentiation'. 4. Addition of GO enrichment analysis of this region of differentiation. Intriguingly, the overall findings from the expanded analysis of the 'island of differentiation' point to regulatory genes rather than hydrocarbon or anthocyanin genes in this region. If true, it may indicate an overlooked role of changes in regulation as a major contributor to genetic speciation. Clearly, this possibility warrants detail investigation in future studies.

Response: Thank you very much for your assessment, with which we certainly fully agree. We hope this study will be published soon and that it will fuel follow-up research to clarify the open questions.

Minor suggestions.

Ln 110-118

The comparisons of estimated percentages of LTR elements are reported to 2 decimal places. However, given that any published genome is never 100% complete, and given some uncertainty in the annotation pipelines, I suggest that it is will never be possible to estimate the percentage of LTR elements to this 2 decimal place level of precision. Surely an integer estimate is even optimistic?

Response: This is a fair point. We followed this suggestion and changed to these numbers to integer estimates.

Elsewhere, I note some 2 decimal place estimates, e.g. Line 100 with 90.01% etc, which should be simplified to integer or at most 1 decimal place.

Response: We followed this suggestion and changed this to 1 decimal place.

Ln 166. Insert 'has' i.e. It has originated several times independently ...

Response: We amended the text as suggested.

Rod Peakall
The Australian National University

Reviewer #3 (Remarks to the Author):

I have now reviewed the revised manuscript "Genome of the early spider-orchid *Ophrys sphegodes* provides insights into sexual deception and pollinator adaptation".

The authors have done a good job of addressing my concerns, as well as that of the other reviewers, which makes for a more comprehensive and balanced paper.

The images are informative and clear, especially the inclusion of the new figure on the hydrocarbon biosynthesis pathway (Figure 3).

The larger field of sexual deception is also acknowledged and briefly discussed, including how sexually deceptive flowers function and are pollinated. Inclusion of more ecological discussion would have been preferable, but as this is primarily a genomic study. I think this is fair.

The authors have contributed significant data that will hopefully open the door to more comparative analyses as more genomes of orchids with different pollination strategies become available.

I think this paper will be of interest to readers and is a valuable contribution to the field.

Response: Thank you for your assessment and for your contribution towards making this a better paper.

Other changes to the manuscript

- To be consistent with the author guidelines, we named and rearranged the order of figure panels for Fig. 3 and Suppl. Fig. 13 so as to match the legend. The legend now refers to panels *a*, *b* and *c*.
- The visual rendering of *Ophrys* *DFR* expression in Suppl. Fig. 13 was inaccurate and therefore corrected.

Reviewers' Comments:

Reviewer #1:

Remarks to the Author:

All the comments have been complete responded (Owing to the manuscript has been published as pre-print, we would like to the manuscript could be more strong), I am glad to suggest this manuscript could be published to Nature Communications.

REVIEWERS' COMMENTS

Reviewer #1 (Remarks to the Author):

All the comments have been complete responded (Owing to the manuscript has been published as pre-print, we would like to the manuscript could be more strong), I am glad to suggest this manuscript could be published to Nature Communications.

Response:

We are delighted to hear Reviewer #1 is now satisfied with the manuscript. We thank Reviewer #1 for having helped to improve this manuscript.